# HyperDPO: Conditioned One-Shot Multi-Objective Fine-Tuning Framework

## Abstract

In LLM alignment and many other ML applications, one often faces the *Multi-Objective Fine-Tuning (MOFT)* problem, *i.e.* fine-tuning an existing model with datasets labeled w.r.t. different objectives simultaneously. To address the challenge, we propose the *HyperDPO* framework, a conditioned one-shot fine-tuning approach that extends the Direct Preference Optimization (DPO) technique, originally developed for efficient LLM alignment with preference data, to accommodate the MOFT settings. By substituting the Bradley-Terry-Luce model in DPO with the Plackett-Luce model, our framework is capable of handling a wide range of MOFT tasks that involve listwise ranking datasets. Compared with previous approaches, HyperDPO enjoys an efficient one-shot training process for profiling the Pareto front of auxiliary objectives, and offers post-training control over trade-offs. Additionally, we propose a novel *Hyper Prompt Tuning* design, that conveys continuous importance weight across objectives to transformer-based models without altering their architecture, and investigate the potential of *temperature-conditioned networks* for enhancing the flexibility of post-training control. We demonstrate the effectiveness and efficiency of the HyperDPO framework through its applications to various tasks, including Learning-to-Rank (LTR) and LLM alignment, highlighting its viability for large-scale ML deployments.

## 1 Introduction

*Direct Preference Optimization (DPO)* (Rafailov et al., 2024b) has been introduced as a memory- and computation-efficient alternative to the traditional *Reinforcement Learning with Human Feedback (RLHF)* (Christiano et al., 2017; Stiennon et al., 2020; Ouyang et al., 2022) in Large Language Model (LLM) alignment. The method fine-tunes a pre-trained LLM with additional data that indicates the preference between different proposals w.r.t. customized objectives, such as safety, verbosity, coherence, *etc.* (Wu et al., 2024b). The idea of DPO is to reparametrize the *reward function* in RLHF and guide the fine-tuning process in a supervised learning manner with the preference data.

LLM alignment also intersects with the *Multi-Objective Optimization (MOO)* problem, which involves fine-tuning a model w.r.t. multiple objectives simultaneously (Ji et al., 2024b; Wu et al., 2024b; Zhou et al., 2023; Rame et al., 2024). In many MOO scenarios within machine learning, a pre-existing model optimized for one or more *main objectives* is further aligned to a set of *auxiliary objectives* without significantly detracting the model's performance on the main objectives in order to achieve certain desirable properties (Navon et al., 2020; Ruchte & Grabocka, 2021). This specific scenario is termed the *Multi-Objective Fine-Tuning (MOFT)* problem. As auxiliary objectives may conflict with each other, the notion of alignment is generalized to achieving the *Pareto optimality* in the MOFT setting, where the goal is to profile the *Pareto front*, representing a spectrum of trade-off solutions where no single auxiliary objective can be improved without compromising another.

In this work, we address the task of multi-objective fine-tuning in a broad context through our proposed *HyperDPO* framework. This conditioned one-shot multi-objective fine-tuning framework is designed to (1) generalize DPO to the MOFT setting, (2) profile the Pareto front of the auxiliary objectives while maintaining the model performance on the main objectives with an efficient one-shot training process, and (3) offer as flexible post-training controls over the trade-offs as possible.

### 1.1 Contributions

The main contributions of this work are as follows:

- We propose the *HyperDPO* method, a conditioned one-shot multi-objective fine-tuning framework that generalizes DPO to the multi-objective setting, profiles the Pareto front through one-shot training, and offers flexible post-training control over trade-offs.
- The HyperDPO framework is tested across diverse tasks, including Learning-to-Rank (LTR) and LLM alignment tasks, demonstrating its state-of-the-art performance to achieve comprehensive Pareto fronts against existing baselines and its efficiency across a wide range of high-dimensional, multi-objective, large-scale applications.
- For LLM applications, we develop a novel *Hyper Prompt Tuning* design that translates the continuous importance weight into a mask applied to the prefix embedding, effectively conveying weights across auxiliary objectives to the LLM without altering its underlying architecture.
- We further investigate the potential of the *temperature-conditioned network* for enhancing the flexibility of post-training control over the trade-offs, promising broader application of the HyperDPO framework to more complex multi-objective fine-tuning scenarios.

## 1.2 RELATED WORKS

**LLM Alignment.** LLM alignment has been a popular topic in the machine learning community. Reinforcement Learning from Human Feedback (RLHF) has been a groundbreaking technique for alignment (Christiano et al., 2017; Schulman et al., 2017; Ouyang et al., 2022; Bai et al., 2022a), which serves as a foundation for training models like GPT-4 (Achiam et al., 2023), and several advances have been made in this direction (Dong et al., 2024; Bai et al., 2022b; Lee et al., 2023). To reduce computational complexity, Direct Preference Optimization (DPO) (Rafailov et al., 2024b) has been proposed as an alternative to RLHF, and further developed in the literature (Pal et al., 2024; Wu et al., 2024a; Gheshlaghi Azar et al., 2023; Tang et al., 2024b; Rafailov et al., 2024a; Zeng et al., 2024; Liu et al., 2024; Song et al., 2024; Zhou et al., 2023; Guo et al., 2024; Yang et al., 2024). We refer readers to Shen et al. (2023); Wang et al. (2024) for comprehensive reviews on LLM alignment.

**Multi-Objective Optimization.** Multi-Objective Optimization (MOO) has been actively studied in control systems (Gambier & Badreddin, 2007) and economics (Tapia & Coello, 2007). The main focus of the related research is the development of algorithms to profile Pareto fronts efficiently so as to understand the trade-offs between objectives. Traditional methods include the evolutionary algorithms (Zhou et al., 2011) and Bayesian optimization (Laumanns & Ocenasek, 2002). Recently, gradient-based MOO methods have been studied in the machine learning settings (Sener & Koltun, 2018; Lin et al., 2019; Mahapatra & Rajan, 2020; Liu & Vicente, 2021; Ren et al., 2024). Hypernetwork-based methods are also explored by a series of works (Navon et al., 2020; Lin et al., 2020; Chen & Kwok, 2022; Hoang et al., 2023).

**Learning-to-Rank (LTR).** Learning to Rank (LTR) (Liu et al., 2009) tasks differ from traditional supervised learning in that they do not associate each sample with a simple label; instead, an optimal order of items within a group to maximize metrics, *e.g.* Normalized Discount Cumulative Gain (NDCG) (Järvelin & Kekäläinen, 2002; Wang et al., 2013). Typically, LTR models score documents and rank them thereby. To bridge LTR with supervised learning, various differentiable losses have been proposed as the proxy to these metrics (Burges et al., 2006; Taylor et al., 2008; Cao et al., 2007; Qin et al., 2021; Swezey et al., 2021). In the context of Multi-Objective LTR, existing work includes label aggregation (Dai et al., 2011; Carmel et al., 2020), loss aggregation (Hu & Li, 2018; Mahapatra et al., 2023a;b; Tang et al., 2024a), and hypernetwork (Chen et al., 2023).

## 2 PRELIMINARIES

In this section, we briefly introduce the proximal and direct preference optimization frameworks for fine-tuning LLMs with preference data, the MOO problem in machine learning settings, and related definitions.

### 2.1 PROXIMAL AND DIRECT PREFERENCE OPTIMIZATION

Suppose we have a base LLM $p_{\text{base}}(y|\boldsymbol{x})$, with $\boldsymbol{x}$ and $y$ being the content and the proposal, respectively, and $p_{\text{base}}(y|\boldsymbol{x})$ the probability of generating response $y$ given $\boldsymbol{x}$. The goal of DPO is to fine-tune the model $p_{\text{base}}(y|\boldsymbol{x})$ with the preference data $\mathcal{D}_{\text{DPO}} = \{(\boldsymbol{x}^{(k)}, y_1^{(k)} > y_2^{(k)})\}_{k \in [N]}$, where $y_1^{(k)} > y_2^{(k)}$ denotes $y_1^{(k)}$ is more preferred than $y_2^{(k)}$ in the context of $\boldsymbol{x}^{(k)}$.

**Proximal Preference Optimization.** In RLHF (Christiano et al., 2017) or *Proximal Preference Optimization (PPO)* (Schulman et al., 2017), one first models the preference data by the *Bradley-Terry-Luce (BTL) model* (Bradley & Terry, 1952):

$$\mathbb{P}(y_1 > y_2 | \boldsymbol{x}) = \frac{\exp(r(y_1|\boldsymbol{x}))}{\exp(r(y_1|\boldsymbol{x})) + \exp(r(y_2|\boldsymbol{x}))} = \sigma\left(r(y_1|\boldsymbol{x}) - r(y_2|\boldsymbol{x})\right), \tag{1}$$

where $r(y|\boldsymbol{x})$ is the reward function and $\sigma(\cdot)$ is the sigmoid function. PPO is carried out in the following two steps:

*Step 1.* Parametrize $r(y|\boldsymbol{x})$ by a neural network $r_\phi(y|\boldsymbol{x})$, where the parameters $\phi$ are trained by maximizing the log-likelihood of the preference data:

$$-\mathcal{L}(r_\phi; \mathcal{D}_{\mathrm{DPO}}) = \mathbb{E}_{(\boldsymbol{x}, y_1 > y_2)}\left[\log \sigma(r_\phi(y_1|\boldsymbol{x}) - r_\phi(y_2|\boldsymbol{x}))\right]; \tag{2}$$

*Step 2.* Fine-tune the base model $p_{\mathrm{base}}(y|\boldsymbol{x})$ by maximizing the expected reward with respect to the preference data while maintaining the KL-divergence between the refined model and the base model:

$$-\mathcal{L}(p_\theta; p_{\mathrm{base}}, r_\phi, \beta) = \mathbb{E}\left[r_\phi(y|\boldsymbol{x})\right] - \beta D_{\mathrm{KL}}(p_\theta || p_{\mathrm{base}}) = \mathbb{E}\left[r_\phi(y|\boldsymbol{x}) - \beta \log \frac{p_\theta(y|\boldsymbol{x})}{p_{\mathrm{base}}(y|\boldsymbol{x})}\right], \tag{3}$$

where $\beta > 0$ is called the *temperature parameter* that controls the scale of the fine-tuning.

**Direct Preference Optimization.** The observation that motivates DPO (Rafailov et al., 2024b) is that the reward function $r_\phi(\boldsymbol{x}, y)$ in (3) can be solved explicitly by letting $r_\theta(y|\boldsymbol{x}) = \beta \log \frac{p_\theta(y|\boldsymbol{x})}{p_{\mathrm{base}}(y|\boldsymbol{x})}$, and therefore, the training process can be simplified to a one-shot logistic regression problem:

$$-\mathcal{L}(p_\theta; p_{\mathrm{base}}, \beta, \mathcal{D}_{\mathrm{DPO}}) = \mathbb{E}_{(\boldsymbol{x}, y_1 > y_2)}\left[\log \sigma\left(\beta \log \frac{p_\theta(y_1|\boldsymbol{x})}{p_{\mathrm{base}}(y_1|\boldsymbol{x})} - \beta \log \frac{p_\theta(y_2|\boldsymbol{x})}{p_{\mathrm{base}}(y_2|\boldsymbol{x})}\right)\right]. \tag{4}$$

For completeness, we provide the proofs of the claim above in Appendix A.1.

## 2.2 Multi-Objective Optimization

In contrast to its single-objective counterpart, MOO considers the optimization problem with multiple objectives $\min_{\theta \in \Theta} \boldsymbol{\mathcal{L}}(\theta) = (\mathcal{L}_1(\theta), \mathcal{L}_2(\theta), \ldots, \mathcal{L}_m(\theta))$, where $\Theta$ is the feasible region. The goal is to profile the Pareto front, which is defined as follows:

$$\mathcal{P} = \{\theta \in \Theta : \nexists \theta' \in \Theta \text{ s.t. } \forall i \in [m], \mathcal{L}_i(\theta') \leqslant \mathcal{L}_i(\theta) \text{ and } \exists j \in [m], \mathcal{L}_j(\theta') < \mathcal{L}_j(\theta)\},$$

intuitively translating to the set of trade-off solutions that cannot be improved in one without worsening another. This concept is motivated by the possible conflicts between the objectives, and one may observe the details of the trade-offs from the Pareto front and make informed decisions accordingly.

For many machine learning applications, the MOO problem can be formulated as follows: given a dataset in the form of $\mathcal{D}_{\mathrm{MOO}} = \{\mathcal{D}_{\mathrm{MOO}}^j\}_{j \in [m]} = \{\{\boldsymbol{y}^{(k)}, z^{j,(k)}\}_{k \in [N]}\}_{j \in [m]}$, where $\boldsymbol{y}^{(k)}$ is the feature vector and $z^{j,(k)}$ is the $j$-th label of the $k$-th data point, the goal is to learn a model $f_\theta(\boldsymbol{y})$ that optimizes the following objectives:

$$\min_{\theta \in \Theta} \boldsymbol{\mathcal{L}}(f_\theta; \mathcal{D}_{\mathrm{MOO}}) := (\mathcal{L}_1(f_\theta; \mathcal{D}_{\mathrm{MOO}}^1), \mathcal{L}_2(f_\theta; \mathcal{D}_{\mathrm{MOO}}^2), \ldots, \mathcal{L}_m(f_\theta; \mathcal{D}_{\mathrm{MOO}}^m)), \tag{5}$$

where $\mathcal{L}_j(f_\theta; \mathcal{D}_{\mathrm{MOO}}^j)$ is the loss function for the model $f_\theta$ with respect to the $j$-th objective, and the feasible region $\Theta$ is over all possible model parameters.

## 3 Methodology

In this section, we first introduce the multi-objective fine-tuning problem and its relation to the LLM alignment problem. Then, we present the HyperDPO framework, a conditioned one-shot multi-objective fine-tuning framework that generalizes the DPO framework to the MOFT setting and profiles the Pareto front of the auxiliary objectives.

## 3.1 MULTI-OBJECTIVE FINE-TUNING

The MOFT problem is a generalization of the LLM alignment problem to the multi-objective setting, where the goal is to fine-tune an existing base model $p_{\text{base}}(y|\boldsymbol{x})$ with respect to multiple *auxiliary* objectives simultaneously while maintaining the model performance on the *main* objective(s) that the base model is optimized for.

In this work, we formulate the MOFT problem as follows: given a set of item groups, each of which contains a list of items and corresponding labels with respect to $m$ different objectives. The dataset is in the form of

$$\mathcal{D}_{\text{MOFT}} = \{\mathcal{D}_{\text{MOFT}}^j\}_{j\in[m]} = \left\{ \left\{ \boldsymbol{x}^{(k)}, (\boldsymbol{y}_i^{(k)})_{i\in[n^{(k)}]}, (z_i^{j,(k)})_{i\in[n^{(k)}]} \right\}_{k\in[N]} \right\}_{j\in[m]}, \tag{6}$$

where $n^{(k)}$ is the number of items, $\boldsymbol{x}^{(k)} \in \mathbb{R}^D$ denotes the context and $\boldsymbol{y}_i^{(k)} \in \mathbb{R}^d$ denotes the feature vector of the $i$-th item, and $z_i^{j,(k)} \in \mathbb{R}^{n^{(k)}}$ denotes the $j$-th label of the $i$-th item, in the $k$-th item group, which often indicates the preference tendency of each item with respect to the $j$-th aspect.

**Relation to the Learning-to-Rank Task.** Datasets in this particular form are closely related to the Learning-to-Rank (LTR) problem, as one may immediately derive a ranking of the items in each group by sorting with respect to the labels $z_i^{j,(k)}$. In general, the dataset (6) may contain not only $\binom{n}{2}$ pairwise preference data but also the comparative intensity of the preferences, necessitating generalized models to handle the MOFT task. The LTR task will be discussed in more detail in Section 4.1 as we present the application of the HyperDPO framework to it.

**Relation to the LLM Alignment.** The preference dataset $\mathcal{D}_{\text{DPO}}$ in LLM alignment can be viewed as a special case of the MOFT problem, where the number of auxiliary objectives $m = 1$, the number of items (proposals) in each group $n = 2$, and the label $z_i^{1,(k)}$ is binary, being 1 if the $i$-th item is preferred over the other, and 0 otherwise. We also refer to Liu et al. (2024); Song et al. (2024) for more discussions on LLM alignment with listwise data.

**Relation to the MOO task.** MOFT is a generalization of the MOO problem (5) to the fine-tuning setting, where the model $f_\theta(\boldsymbol{y})$ is the new model $p_\theta(\boldsymbol{y}|\boldsymbol{x})$, and the dataset $\mathcal{D}_{\text{MOO}}$ is the preference dataset $\mathcal{D}_{\text{MOFT}}$ (6). The MOFT problem can be formulated in the MOO language as follows:

$$\min_{\theta\in\Theta} \boldsymbol{\mathcal{L}}(p_\theta; p_{\text{base}}, \boldsymbol{\beta}, \mathcal{D}_{\text{MOFT}}) = (\mathcal{L}_j(p_\theta; p_{\text{base}}, \beta_j, \mathcal{D}_{\text{MOFT}}^j))_{j\in[m]}, \tag{7}$$

in which the specific choices of the loss functions will be introduced in the next section.

## 3.2 FROM PREFERENCE TO RANKING

Recall that DPO is obtained by *reparametrizing* the reward function in PPO (3) by the ratio of the model probabilities as in (4), one may generalize the DPO framework from preference to ranking datasets, by switching from the BTL model to the Plackett-Luce (PL) model (*cf.* (1) and (8)), as proposed by Liu et al. (2024). For the clarity of further generalization, we give a brief recapitulation of the PL model and its relation to the ListNet loss (Cao et al., 2007) in the following.

**Plackett-Luce Model.** PL model (Plackett, 1975) is one of the most popular ways to model the ranking data. In the PL model, the probability of a ranking is postulated as:

$$\mathbb{P}(\boldsymbol{y}_{\pi_1} > \boldsymbol{y}_{\pi_2} > \cdots > \boldsymbol{y}_{\pi_n}|\boldsymbol{x}) := \prod_{i=1}^n \frac{\exp(s(\boldsymbol{y}_{\pi_i}|\boldsymbol{x}))}{\sum_{k=i}^n \exp(s(\boldsymbol{y}_{\pi_k}|\boldsymbol{x}))}, \tag{8}$$

where $s(\boldsymbol{y}|\boldsymbol{x})$ is the score function. The model is trained by aligning the $j$-th label with the top-one probability of the PL model $\mathbb{P}(\boldsymbol{y}_i > \boldsymbol{y}_{i'}, \forall i' \neq i|\boldsymbol{x}) = \frac{\exp(s(\boldsymbol{y}_i|\boldsymbol{x}))}{\sum_{i'=1}^n \exp(s(\boldsymbol{y}_{i'}|\boldsymbol{x}))}$, *i.e.* the ListNet loss (Cao et al., 2007):

$$-\mathcal{L}_{\text{ListNet}}(s_\theta; \mathcal{D}_{\text{LTR}}^j) = \mathbb{E}\left[ \sum_{i=1}^n t(z_i^j) \log\left( \frac{\exp(s_\theta(\boldsymbol{y}_i|\boldsymbol{x}))}{\sum_{i'=1}^n \exp(s_\theta(\boldsymbol{y}_{i'}|\boldsymbol{x}))} \right) \right], \tag{9}$$

where the expectation is taken over the data distribution of $\mathcal{D}_{\text{LTR}}$, and $t(\cdot)$ is an appropriate normalization of the label vector $\boldsymbol{z}$ s.t. $\sum_{i=1}^n t(z_i) = 1$. Common choices include the softmax function

for dense labels and $L_1$ normalization for sparse labels, corresponding to different modeling of the ranking data.

The log-likelihood $\log p_\theta(\boldsymbol{y}|\boldsymbol{x})$ is related to the score function $s_\theta(\boldsymbol{y}|\boldsymbol{x})$ by the softmax function, mimicking the BTL model (1) in which $\log p_\theta(\boldsymbol{y}|\boldsymbol{x})$ is related to the reward function $r_\theta(\boldsymbol{y}|\boldsymbol{x})$ by the sigmoid function. Therefore, given the ranking dataset $\mathcal{D}_{\mathrm{MOFT}}$ (6), the loss function (4) of the $j$-th aspect can be modified to, incorporating the ListNet loss (9):

$$-\mathcal{L}_{\mathrm{ListNet}}(s_\theta; s_{\mathrm{base}}, \beta_j, \mathcal{D}_{\mathrm{LTR}}^j) = \mathbb{E}\left[\sum_{i=1}^{n} t(z_i^j) \log\left(\frac{\exp\left(\beta_j(s_\theta(\boldsymbol{y}_i|\boldsymbol{x}) - s_{\mathrm{base}}(\boldsymbol{y}_i|\boldsymbol{x}))\right)}{\sum_{i'=1}^{n} \exp\left(\beta_j(s_\theta(\boldsymbol{y}_{i'}|\boldsymbol{x}) - s_{\mathrm{base}}(\boldsymbol{y}_{i'}|\boldsymbol{x}))\right)}\right)\right].$$
$$(10)$$

For completeness, the proof of this claim is provided in Appendix A.1. One should notice that when $t(\cdot)$ is the $L^1$ normalization, the ListNet loss (9) applied to the preference dataset $\mathcal{D}_{\mathrm{DPO}}$ in the form of binary labels is equivalent to the DPO loss (4).

## 3.3 MOFT WITH IMPORTANCE-CONDITIONED NETWORKS

With the introduction of the ListNet loss (9), we may rewrite the MOFT problem (7) in a more detailed form:

$$\min_{\theta \in \Theta} \boldsymbol{\mathcal{L}}_{\mathrm{ListNet}}(s_\theta; s_{\mathrm{base}}, \boldsymbol{\beta}, \mathcal{D}_{\mathrm{MOFT}}) = (\mathcal{L}_{\mathrm{ListNet}}(s_\theta; s_{\mathrm{base}}, \beta_j, \mathcal{D}_{\mathrm{MOFT}}^j))_{j \in [m]}. \quad (11)$$

We assume the temperature parameter $\boldsymbol{\beta} = (\beta_1, \beta_2, \ldots, \beta_m) \in \mathbb{R}_+^m$ that controls the trade-off between the main objective and each auxiliary objective is fixed for now.

The most straightforward way to solve this MOO problem is to train the model $s_\theta$ with a linear combination of the preference data (Zhou et al., 2023):

$$\mathcal{L}_{\mathrm{ListNet},\boldsymbol{w}}(s_\theta; s_{\mathrm{base}}, \boldsymbol{\beta}, \mathcal{D}_{\mathrm{MOFT}}) := \boldsymbol{w}^\top \boldsymbol{\mathcal{L}}_{\mathrm{ListNet}}(s_\theta; s_{\mathrm{base}}, \boldsymbol{\beta}, \mathcal{D}_{\mathrm{MOFT}}), \quad (12)$$

where $\boldsymbol{w} = (w_1, w_2, \ldots, w_m)^\top \in \Delta^m$ is the weight vector that reflects the importance we assign over the objectives, and with $\Delta^m$ being the $m$-dimensional simplex. As $\boldsymbol{w}$ iterates over $\Delta^m$, the model $s_\theta$ will be optimized over a specific trade-off between the main objective and the auxiliary objectives and possibly land on the Pareto front. This approach is known as the *weighted sum* or *linear scalarization* method in MOO literature and is able to obtain the complete Pareto front when it is convex (Jakob & Blume, 2014).

An efficient way to profile the Pareto front of this MOFT problem is to use *hypernetworks* (Navon et al., 2020; Hoang et al., 2023). As an efficient alternative to traditional hypernetworks that use additional neural networks to generate the model parameters, Ruchte & Grabocka (2021) proposes input-conditioned networks and HyperDPO generalizes this idea to the MOFT settings. To be specific, we propose to design *importance-conditioned* neural networks $s_\theta$ that not only take in the data but also depend on the importance weight $\boldsymbol{w}$ over objectives. Intuitively, it formulates the MOO problem as a "meta-learning" problem, where the model $s_\theta(\cdot, \boldsymbol{w}|\boldsymbol{x})$ is trained to optimize the objectives over a distribution of weight vectors. In practice, in order to foster the exploration of the Pareto front, one may also incorporate artificial penalization terms to the loss function, such as the cosine similarity between the loss vector $\boldsymbol{\mathcal{L}}(s_\theta; s_{\mathrm{base}}, \boldsymbol{\beta}, \mathcal{D}_{\mathrm{MOFT}})$ of the model and the weight vector (Ruchte & Grabocka, 2021):

$$\mathcal{G}_{\boldsymbol{w}}(s_\theta; s_{\mathrm{base}}, \boldsymbol{\beta}) := -\cos\angle\left(\boldsymbol{w}, -\boldsymbol{\mathcal{L}}_{\mathrm{ListNet}}(s_\theta(\cdot, \boldsymbol{w}|\boldsymbol{x}); s_{\mathrm{base}}, \boldsymbol{\beta}, \mathcal{D}_{\mathrm{MOFT}})\right). \quad (13)$$

This penalization term intuitively confines the loss vector $\boldsymbol{\mathcal{L}}_{\mathrm{ListNet}}$ to converging along the direction of the weight vector $\boldsymbol{w}$, empowering possible profiling of concave Pareto fronts (Lin et al., 2019).

We design the following *importance-conditioned one-shot (ICOS) fine-tuning* loss that if optimized, will output models $s_{\theta,\boldsymbol{\beta}}(\cdot, \boldsymbol{w}|\boldsymbol{x})$ that are Pareto optimal with respect to the auxiliary objectives:

$$\mathcal{L}_{\mathrm{ICOS}}(s_\theta; s_{\mathrm{base}}, \boldsymbol{\beta}, \mathcal{D}_{\mathrm{MOFT}}, \boldsymbol{\alpha}, \lambda)$$
$$:= \mathbb{E}_{\boldsymbol{w} \sim \mathrm{Dir}(\boldsymbol{\alpha})}\left[\mathcal{L}_{\mathrm{ListNet},\boldsymbol{w}}(s_\theta(\cdot, \boldsymbol{w}|\boldsymbol{x}); s_{\mathrm{base}}, \boldsymbol{\beta}, \mathcal{D}_{\mathrm{MOFT}}) + \lambda\mathcal{G}_{\boldsymbol{w}}(s_\theta(\cdot, \boldsymbol{w}|\boldsymbol{x}); s_{\mathrm{base}}, \boldsymbol{\beta})\right], \quad (14)$$

where $\boldsymbol{\alpha}$ is the concentration parameter of the Dirichlet distribution over $\Delta^m$, and $\lambda$ is the penalization coefficient. The HyperDPO framework is summarized in Algorithm 1.

## 3.4 LINEAR TRANSFORMATION PROPERTY

---

**Algorithm 1:** HyperDPO Framework

---

**Data:** Base model $s_{\text{base}}(\boldsymbol{y}|\boldsymbol{x})$, dataset $\mathcal{D}_{\text{MOFT}}$, temperature $\boldsymbol{\beta}$, concentration parameter $\boldsymbol{\alpha}$,
   penalization coefficient $\lambda$ (Training); scale $c$, weight vector $\boldsymbol{w}$ (Post-Training Control).
**Result:** Fine-Tuned model $s_{\theta,\cdot\boldsymbol{\beta}}(\cdot,\cdot|\boldsymbol{x})$ (Training); $s_{\theta,c\boldsymbol{\beta}}(\boldsymbol{y},\boldsymbol{w}|\boldsymbol{x})$ (Post-Training Control).
  // Training
 **1 for** $e = 1$ **to** $N_{\text{steps}}$ **do**
 **2**    Sample $\boldsymbol{w}' \sim \text{Dir}(\boldsymbol{\alpha})$;
 **3**    $\theta \leftarrow \theta - \eta \nabla_\theta \left[ \mathcal{L}_{\text{ListNet},\boldsymbol{w}}(s_\theta(\cdot,\boldsymbol{w}'|\boldsymbol{x}); s_{\text{base}}, \boldsymbol{\beta}, \mathcal{D}_{\text{MOFT}}) + \lambda \mathcal{G}_{\boldsymbol{w}}(s_\theta(\cdot,\boldsymbol{w}'|\boldsymbol{x}); s_{\text{base}}, \boldsymbol{\beta}) \right]$;
 **4 end**
  // Post-Training Control
 **5** $s_{\theta,c\boldsymbol{\beta}}(\boldsymbol{y},\boldsymbol{w}|\boldsymbol{x}) \leftarrow (1 - 1/c)\, s_{\text{base}}(\boldsymbol{y}|\boldsymbol{x}) + s_{\theta,\boldsymbol{\beta}}(\boldsymbol{y},\boldsymbol{w}|\boldsymbol{x})/c$.

---

Due to the linearity of the DPO framework, one can show the following linear transformation property:

**Proposition 3.1** (Linear Transformation Property). *For any* $\boldsymbol{\beta} \in \mathbb{R}_+^m$ *and* $\boldsymbol{w} \in \Delta^m$*, we denote the model optimized by the ICOS loss (14) with temperature* $\boldsymbol{\beta}$ *as* $s_{\theta,\boldsymbol{\beta}}(\boldsymbol{y},\boldsymbol{w}|\boldsymbol{x})$*, and suppose the penalization term* $\mathcal{G}_{\boldsymbol{w}}(s_\theta; s_{\text{base}}, \boldsymbol{\beta})$ *is a function of* $\mathcal{L}_{\text{ListNet}}(s_\theta(\cdot,\boldsymbol{w}|\boldsymbol{x}); s_{\text{base}}, \boldsymbol{\beta}, \mathcal{D}_{\text{MOFT}})$*.*

*Then* $s_{\theta,\boldsymbol{\beta}}(\boldsymbol{y},\boldsymbol{w}|\boldsymbol{x})$ *should satisfy the linear transformation that for any* $c > 0$*, we have that*

$$s_{\theta,c\boldsymbol{\beta}}(\boldsymbol{y},\boldsymbol{w}|\boldsymbol{x}) = \left(1 - \frac{1}{c}\right) s_{\text{base}}(\boldsymbol{y}|\boldsymbol{x}) + \frac{1}{c} s_{\theta,\boldsymbol{\beta}}(\boldsymbol{y},\boldsymbol{w}|\boldsymbol{x}) \tag{15}$$

*is also an optimal solution to the ICOS loss (14) with temperature* $c\boldsymbol{\beta}$*.*

The proof of this proposition is provided in Appendix A.2 and will be empirically validated with experiments as shown in Figure 11. Powered by Proposition 3.1, this framework also offers post-training controls

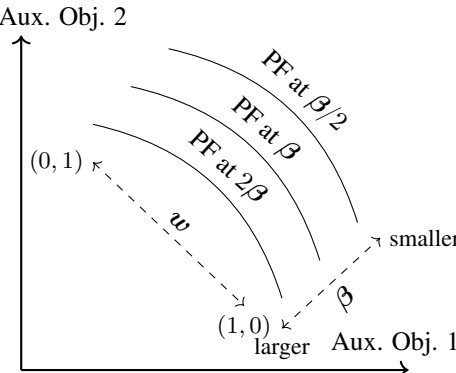

Figure 1: Conceptual Illustration of Available Post-Training Controls in the HyperDPO Framework with 2 auxiliary objectives.

over the trade-offs in the MOFT problem. As illustrated in Figure 1, one may adjust the trade-offs between the auxiliary objectives by adjusting the weight vector $\boldsymbol{w}$, and those between the fidelity to the base model and its performance on the fine-tuning datasets of the new model by scaling the temperature parameter $\boldsymbol{\beta}$ with (15). Furthermore, this property will serve as the foundation for the design of the temperature-conditioned network, which will be discussed in Appendix C.

## 4 EXPERIMENTS

In this section, we provide the detailed experiment design and results of the HyperDPO framework for different applications, including the learning-to-rank task and the LLM alignment task. We also analyze the results and compare them with state-of-the-art methods. The code will be released upon acceptance.

**Baselines.** We compare the HyperDPO framework with the following state-of-the-art baselines:

- *DPO Linear Scalarization (DPO-LS):* We first sample several weight vectors $\boldsymbol{w}$ over the simplex $\Delta^m$ and train the model $s_\theta(\cdot,\boldsymbol{w}|\boldsymbol{x})$ with the weighted sum loss (11). Notably, when $\boldsymbol{w}$ are unit vectors, it returns the result of the single-objective fine-tuning for reference.
- *DPO Soup (Rame et al., 2024):* The DPO Soup method first trains DPO models for each auxiliary objective and then combines the models by a weighted sum.
- *MO-DPO (Zhou et al., 2023):* The MO-DPO method first chooses a weight vector $\boldsymbol{w}$ and then adds a margin reward term depending on $\boldsymbol{w}$ to the DPO loss to ensure multi-objective optimization.

For each baseline, we will use the same number of weight vectors $\boldsymbol{w}$ for a fair comparison. For details and further discussion of these baselines, we refer to Appendix B.1.

**Hypervolume Metric.** We adopt the *hypervolume (HV)* indicator (Zitzler & Künzli, 2004) for evaluating the performance of MOO methods. Assuming the higher evaluation metrics indicate

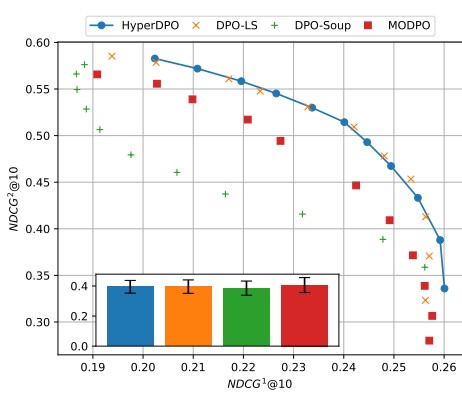 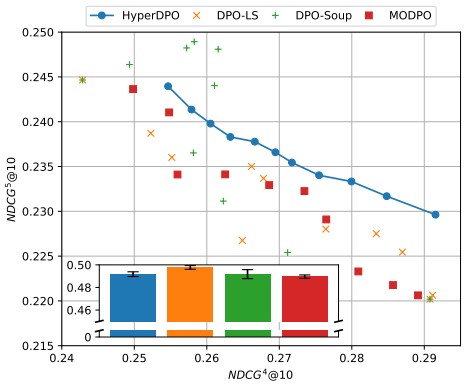

(a) Objective I vs Objective II.    (b) Objective IV vs Objective V.

Figure 2: Comparison of Pareto fronts obtained by the HyperDPO framework and the baselines on the MSLR-WEB10K dataset with 2 auxiliary objectives. Two axes denote the NDCG@10 of the two auxiliary objectives (the higher, the better). The inset plot shows the average NDCG@10 of the main objective, with the error bar denoting the standard deviation across the 11 sampled points.

better performance, the hypervolume of the approximation $\hat{\mathcal{P}}$ to the real Pareto front $\mathcal{P}$ is defined as the volume of the dominated region of $\hat{\mathcal{P}}$ w.r.t. a reference point $\boldsymbol{r}$, *e.g.* when applied to minimization problems, the hypervolume is defined as $\mathrm{HV}(\hat{\mathcal{P}}, \boldsymbol{r}) = \int_{\boldsymbol{x} < \boldsymbol{r}} \mathbf{1}_{\exists \boldsymbol{p} \in \hat{\mathcal{P}}, \boldsymbol{p} \leq \boldsymbol{x}} \mathrm{d}\boldsymbol{x}$. Higher hypervolume values indicate higher quality of the Pareto front.

## 4.1 LEARNING-TO-RANK TASK

We first test the HyperDPO framework on the learning-to-rank task. In this task, $\boldsymbol{x}^{(k)}$ in $\mathcal{D}_{\mathrm{MOFT}}$ denotes a query, and $\boldsymbol{y}_i^{(k)}$ denotes the feature vector of the $i$-th document, and $z_i^{j,(k)}$ denotes the score of the $i$-th document with respect to the $j$-th aspect. The goal is to provide a ranking $\boldsymbol{\pi}$ of the documents with respect to the scores $z_i^{j,(k)}$ for each query $\boldsymbol{x}^{(k)}$, for which the following *Normalized Discounted Cumulative Gain (NDCG)* is used to evaluate its performance:

$$\mathrm{NDCG}^j@\mathrm{k}(\boldsymbol{\pi}) = \mathbb{E}_{(\boldsymbol{x}, \mathrm{y}, \boldsymbol{z}^j)} \left[ \frac{\mathrm{DCG@k}(\boldsymbol{\pi}, \boldsymbol{z}^j)}{\max_{\boldsymbol{\pi}'} \mathrm{DCG@k}(\boldsymbol{\pi}', \boldsymbol{z}^j)} \right], \text{ where } \mathrm{DCG@k}(\boldsymbol{\pi}, \boldsymbol{z}^j) = \sum_{i=1}^{k} \frac{z_{\pi_i}^j}{\log_2(i+1)}.$$

**NN architecture.** As the common practice in the LTR tasks, the information of the query $\boldsymbol{x}$ has often been incorporated into the feature vectors $\boldsymbol{y}_i$ by concatenation or other methods during the upstream data processing. We use a 2-layer transformer architecture of hidden dimension 128 for the base model $s_{\mathrm{base}}(\boldsymbol{y})$, and the model $s_\theta(\cdot, \boldsymbol{w})$ is designed as a 2-layer transformer architecture of hidden dimension 64 with the weight vector $\boldsymbol{w}$ concatenated to the input of the first layer.

**Dataset.** We adopt the Microsoft Learning-to-Rank Web Search (MSLR-WEB10K) dataset (Qin & Liu, 2013) for the LTR task. The MSLR-WEB10K dataset consists of 10,000 groups ($N = 10^4$), each containing a list of webpages retrieved by the search engine in response to the query $\boldsymbol{x}^{(k)}$ and the corresponding features extracted from the webpage. Following the practice by Mahapatra et al. (2023b), we treat the first 131 features as the feature vector ($\boldsymbol{y}_i^{(k)} \in \mathbb{R}^{131}$). We also identify the relevance label $\in [0:4]$ as the main objective used to train the base model, and the last 5 features, *viz.* (I) Query-URL Click Count, (II) URL Click Count, (III) URL Dwell Time, (IV) Quality Score 1, (V) Quality Score 2, with the relevance label, as 5 different auxiliary objectives ($m = 5$) for fine-tuning. The dataset is split into training (60%), validation (20%), and test (20%) datasets, and all results shown below are on the test split.

**Experiment Results.** We first apply the HyperDPO framework to the case where we only have 2 auxiliary objectives ($m = 2$) for better visualization. The results are shown in Figure 2, in which Figure 2a presents the Pareto front of two sparse labels ($t(\boldsymbol{z}) = \boldsymbol{z}/\|\boldsymbol{z}\|_1$ in (9)) with a relatively easy-to-learn convex Pareto front, while Figure 2b presents the Pareto front of two dense labels ($t(\boldsymbol{z}) = \mathrm{softmax}(\boldsymbol{z})$ in (9)) with a more ill-posed Pareto front. HyperDPO obtains comprehensive

| Method | Aux. HV | Avg. Main Score ($\pm$Std) | Training Time (s) | # Parameters |
|---|---|---|---|---|
| DPO-LS | $1.648 \times 10^{-3}$ | 0.3553 ($\pm$ 0.0290) | 14649.15 | 551,232 |
| DPO Soup | $1.468 \times 10^{-3}$ | 0.3823 ($\pm$ 0.0317) | 6061.69 | 250,615 |
| MO-DPO | $1.263 \times 10^{-3}$ | 0.3595 ($\pm$ 0.0242) | 27059.70 | 801,792 |
| **HyperDPO** | $\mathbf{2.039} \times 10^{-3}$ | **0.4320** ($\pm$0.0277) | **4043.47** | **50,432** |

Table 1: Hypervolume metric and training time of HyperDPO and the baselines on the MSLR-WEB10K dataset with 5 auxiliary objectives. The reference point is $(0,0)$, and 11 points are produced for hypervolume calculation. The main score refers to the NDCG@10 of the main objective.

and competitive Pareto fronts that dominate those of the baselines in both pairs of objectives. Notably, HyperDPO is able to obtain a smooth Pareto front in Figure 2b while the baselines fail to do so. With a common temperature parameter $\beta$ used across all methods, the inset plots demonstrate that the superior performance of the HyperDPO framework is not at the cost of the main objective, as the NDCG@10 of the main objective is comparable or even slightly better to some of the baselines.

We also test the HyperDPO framework on a more complicated case where we have 5 auxiliary objectives ($m = 5$). Our results demonstrate our HyperDPO framework is able to achieve a higher hypervolume metric with significantly less training time and number of parameters compared to the baselines and comparably good preservation of the performance on the main objective. While the computational cost of traditional methods, such as DPO-LS and MO-DPO, grows exponentially with the number of objectives, HyperDPO models are able to maintain a linear growth with almost intact performance, indicating the efficiency and capability of the HyperDPO framework in handling high-dimensional MOFT problems in the LTR task.

**Remark on the Training Time.** The training time in Table 1 refers to the duration of all training jobs required for computing the 11-point Pareto front. As described in Algorithm 1, in each epoch during the HyperDPO training, we first sample a single weight vector $\boldsymbol{w}$ and then compute the ICOS loss $\mathcal{L}_{\mathrm{ICOS}}$ and back-propagate the gradients. Therefore, the training does not introduce additional computational cost compared to the training w.r.t. a single objective. However, HyperDPO may require more training epochs to converge due to the exploration of the Pareto front. In practice, we find that the HyperDPO framework converges rapidly, and the training time may only be slightly longer than that of a single model training.

**Ablation Studies.** We provide the ablation studies of the HyperDPO framework on the LTR task in Appendix B.2. Specifically, we evaluate the sensitivity of the HyperDPO framework to the concentration parameter $\boldsymbol{\alpha}$ (*cf.* Appendix B.2.1) and the model depth (capacity) (*cf.* Appendix B.2.2). Furthermore, we will introduce, discuss the suitability, and compare the performance of two different NN parametrizations of $s_\theta(\cdot, \boldsymbol{w}|\boldsymbol{x})$ in Appendix B.2.3, namely (a) *Training from Scratch* and (b) *Augmentation Network*, which exhibit different trade-offs between the performance and the computational cost and thus may serve different purposes in practice.

As mentioned in Section 3.4, besides the weight vector $\boldsymbol{w}$, the HyperDPO framework also offers post-training control over the temperature parameter $\boldsymbol{\beta}$ via the linear transformation property (Proposition 3.1). We provide examples of the post-training control that is consistent with Figure 1 in Appendix C.1. However, the linear transformation property only offers proportional scaling of the temperature parameter $\boldsymbol{\beta}$, motivating the design and development of the more sophisticated *temperature-conditioned network*. Intuitively, the temperature-conditioned network further incorporate the information of the temperature parameter $\boldsymbol{\beta}$ into the model $s_\theta(\cdot, \boldsymbol{w}, \boldsymbol{\beta}|\boldsymbol{x})$, allowing for more flexible and versatile post-training control. The details of our approach and some preliminary results are presented in Appendix C.

## 4.2 LLM Alignment Task

We then apply the HyperDPO framework to the LLM alignment task. In this task, $\boldsymbol{x}^{(k)}$ in $\mathcal{D}_{\mathrm{MOFT}}$ denotes a prompt, and $\boldsymbol{y}_i^{(k)}$ denotes the response generated by the LLM, and $z_i^{j,(k)}$ denotes the score of the $i$-th response with respect to the $j$-th aspect. The goal is to align the LLM to generate responses that satisfy the auxiliary objectives (*e.g.* verboseness, harmlessness, *etc.*) while maintaining its performance on general tasks (*e.g.* fluency, relevance, *etc.*).

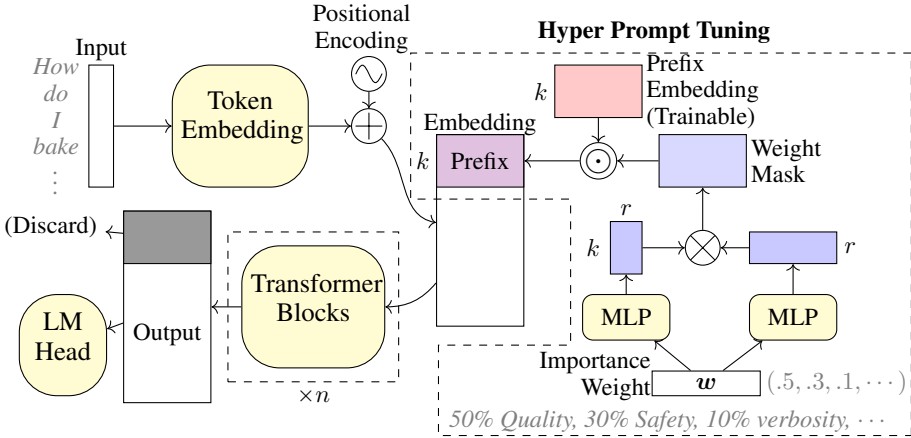

Figure 3: Illustration of the implementation of the HyperDPO framework for the LLM alignment task. The proposed **Hyper Prompt Tuning** method, highlighted within the dashed box on the right, transforms the importance weight vector $\boldsymbol{w}$ into a weight mask and passes it to the LLM via prompt tuning. $k$ denotes the number of virtual tokens, and $r$ is the rank of the weight mask.

**Model Implementation.** In contrast to the LTR task, where we directly concatenate the weight vector $\boldsymbol{w}$ to the input of the model $s_\theta(\cdot, \boldsymbol{w}|\boldsymbol{x})$ in our importance-conditioned network design. In the LLM alignment task, this strategy is generally infeasible, since the input of transformers is tokenized prompts, and the weight vector $\boldsymbol{w}$ is a real vector that is not considered in tokenization and thus a direct concatenation may cause the model to fail to generate meaningful responses.

To address this issue and incorporate the information of the weight vector $\boldsymbol{w}$ into the LLM with the least modification to the model and the training process, we propose a novel design, called *Hyper Prompt Tuning (HPT)*. The mechanism of HPT is shown in Figure 3. Inspired by Prompt Tuning (Lester et al., 2021; Wang et al., 2023), HPT augments the input embedding obtained post token embedding and positional encoding with a trainable prefix embedding block that is controlled by the weight vector $\boldsymbol{w}$. Specifically, HPT follows the following steps:

*Step 1.* HPT takes in a weight vector $\boldsymbol{w} \in \Delta^m$ that indicates the importance across additional objectives and, through two simple trainable MLPs, produces two matrices, the matrix product of which forms the weight mask;

*Step 2.* The weight mask is multiplied entrywise with a trainable prefix embedding block with $k$ virtual tokens;

*Step 3.* The prefix embedding block is then concatenated to the input embedding as a prefix and fed into the transformer blocks of the LLM.

In contrast to Multi-Task Prompt Tuning (Wang et al., 2023), which can only handle a finite number of tasks, one can pass a wide spectrum of importance information by HPT into the LLM, offering flexibility and versatility for our importance-conditioned one-shot fine-tuning implementation. Our implementation of HPT is compatible with the PEFT package and does not depend on any specific LLM architecture.

**Dataset.** The PKU-SafeRLHF dataset (Ji et al., 2024a) is adopted for experiments, which consists of 83.4k entries, each containing a prompt and a pair of responses ($n = 2$) annotated with preferences with respect to both harmlessness and helpfulness ($m = 2$). When the $k$-th response is annotated as more helpful, we assign $z_1^{(k)} = 1$; otherwise, $z_1^{(k)} = 0$. Similarly, when the $k$-th response is annotated as more harmless, we assign $z_2^{(k)} = 1$; otherwise, $z_2^{(k)} = 0$. The goal is to fine-tune the model to generate responses that are both harmless and helpful as a multi-objective optimization problem.

**Training Settings.** We perform fine-tuning to the GPT-2 model (Radford et al., 2019) and the Alpaca-7B-Reproduced model (Dai et al., 2023), following the practice by Zhou et al. (2023) via Parameter-Efficient Fine-Tining (PEFT) with $\alpha = 8$ and $r = 4$ in the low-rank adaptions (LoRA) to the modules within the model. For HyperDPO, we adopt the Hyper Prompt Tuning technique with $k = 8$ and $r = 4$. To ensure a fair comparison, baseline methods will also be augmented with

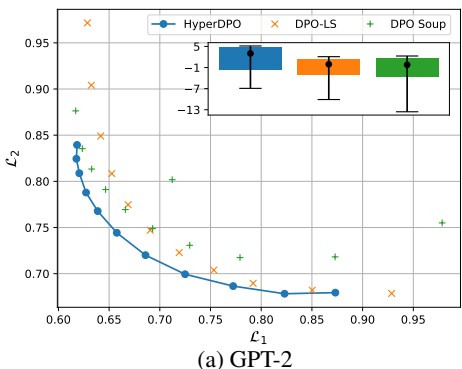

(a) GPT-2

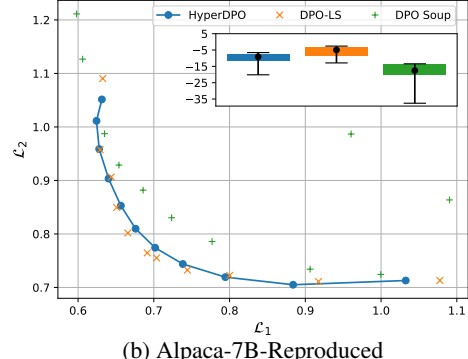

(b) Alpaca-7B-Reproduced

Figure 4: Comparison of Pareto fronts obtained by the HyperDPO framework and the baselines on the PKU-SafeRLHF dataset. Two axes denote the expected cross entropy error of the two auxiliary objectives (the lower, the better). The inset plot shows the interquartile range (IQR) of the deviation of the log-likelihood of the response from the reference model across the test dataset.

| Method | GPT-2 | | Alpaca-7B-Reproduced | |
|---|---|---|---|---|
| | HV | Training Time (s) | HV | Training Time (s) |
| DPO-LS | 0.17668 | 15148.53 | 0.16873 | 94156.12 |
| DPO Soup | 0.18401 | 2755.51 | 0.14270 | 17138.74 |
| **HyperDPO** | **0.19424** | **1396.81** | **0.16885** | **8520.17** |

Table 2: Hypervolume metric and training time of HyperDPO and the baselines on the PKU-SafeRLHF dataset. The reference point for the hypervolume metric is set to $(1.1, 1.1)$, and 11 points are produced for the hypervolume calculation.

the prompt tuning of $k = 8$ on top of LoRA. The HyperDPO framework is built upon the TRL package (von Werra et al., 2020), and the implementation of the HPT is compatible with the PEFT package (Mangrulkar et al., 2022), which allows for easy integration with existing LLMs. All the experiments are conducted on a cluster with $8\times$ NVIDIA A100 GPUs.

**Experiment Results.** In this task, we compare the results of HyperDPO with those of DPO-LS and DPO Soup and we refer readers to discussions in Appendix B.1 for the comparison with MO-DPO. For all experiments, we have chosen a common temperature $\beta = 0.1$ to balance the trade-offs between the main and auxiliary objectives. HyperDPO achieves smooth and comprehensive Pareto fronts (*cf.* Figure 4) with higher hypervolume metrics and less training time (*cf.* Table 2) for both LLM architectures compared to the baselines, demonstrating the effectiveness of the Hyper-DPO framework in the large-scale LLM alignment tasks. Notably, as HyperDPO tackles a "meta-learning" problem that is intrinsically more challenging and thus demands more expressive power, the HyperDPO framework is less prone to overfitting and more robust to the choice of the hyperparameters compared to the baselines. Several ablation studies are provided in Appendix B.2.

## 5 DISCUSSION

In this work, we propose the HyperDPO framework for multi-objective fine-tuning, which is inspired by the DPO framework to profile the Pareto front for a wide range of multi-objective fine-tuning problems with a conditioned one-shot fine-tuning approach. Our method presented superior performance in both the LTR and the large-scale LLM alignment tasks with multiple auxiliary objectives compared to the state-of-the-art methods, demonstrating the effectiveness and efficiency of the HyperDPO framework in handling high-dimensional MOFT problems. Our newly proposed Hyper Prompt Tuning technique also provides a novel way to incorporate importance information into the LLM, offering flexibility for both the importance-conditioned implementation and further research in LLM alignment. We also explored the possibility of temperature-conditioned networks in supplementary materials, opening up new directions for future research. Our work has proven the potential of the HyperDPO framework, and we expect it to incorporate other possible MOO techniques and be further explored in various multi-objective fine-tuning problems in the future.

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

## A  Missing Proofs

In this section, we provide the proofs of the propositions and theorems mentioned in the main text.

### A.1  Proofs of Reparametrization-Related Arguments

*Proof of (4).* Recall that in the second step of PPO, we consider the loss function (3) as follows:

$$-\mathcal{L}(p_\theta; p_{\text{base}}, r_\phi, \beta) = \mathbb{E}_{(\boldsymbol{x}, y)}\left[r_\phi(y|\boldsymbol{x}) - \beta \log \frac{p_\theta(y|\boldsymbol{x})}{p_{\text{base}}(y|\boldsymbol{x})}\right]$$

$$= \int \left(r_\phi(y|\boldsymbol{x}) - \beta \log \frac{p_\theta(y|\boldsymbol{x})}{p_{\text{base}}(y|\boldsymbol{x})}\right) p_\theta(y|\boldsymbol{x}) \mathrm{d}y,$$

we calculate the functional derivative of the loss w.r.t. the density function $p_\theta(y|\boldsymbol{x})$:

$$\frac{\delta \mathcal{L}(p_\theta; p_{\text{base}}, r_\phi, \beta)}{\delta p_\theta(y|\boldsymbol{x})} = \lim_{\epsilon \to 0} \frac{\mathcal{L}(p_\theta + \epsilon \delta p_\theta; p_{\text{base}}, r_\phi, \beta) - \mathcal{L}(p_\theta; p_{\text{base}}, r_\phi, \beta)}{\epsilon}$$

$$= \lim_{\epsilon \to 0} \frac{1}{\epsilon}\left[\int \left(r_\phi(y|\boldsymbol{x}) - \beta \log \frac{p_\theta(y|\boldsymbol{x})}{p_{\text{base}}(y|\boldsymbol{x})} - \beta \frac{\epsilon \delta p_\theta(y|\boldsymbol{x})}{p_\theta(y|\boldsymbol{x})}\right)(p_\theta(y|\boldsymbol{x}) + \epsilon \delta p_\theta(y|\boldsymbol{x})) \mathrm{d}y\right.$$

$$\left. - \int \left(r_\phi(y|\boldsymbol{x}) - \beta \log \frac{p_\theta(y|\boldsymbol{x})}{p_{\text{base}}(y|\boldsymbol{x})}\right) p_\theta(y|\boldsymbol{x}) \mathrm{d}y\right]$$

$$= \int \left(r_\phi(y|\boldsymbol{x}) - \beta \log \frac{p_\theta(y|\boldsymbol{x})}{p_{\text{base}}(y|\boldsymbol{x})} - \beta\right) \delta p_\theta(y|\boldsymbol{x}) \mathrm{d}y.$$

Let the functional derivative vanish, we obtain

$$r_\phi(y|\boldsymbol{x}) = \beta \log \frac{p_\theta(y|\boldsymbol{x})}{p_{\text{base}}(y|\boldsymbol{x})} + \beta,$$

*i.e.*

$$p_\theta(y|\boldsymbol{x}) \propto p_{\text{base}}(y|\boldsymbol{x}) \exp\left(\frac{r_\phi(y|\boldsymbol{x})}{\beta}\right).$$

Since the likelihood $\mathbb{P}(y_1 > y_2|\boldsymbol{x})$ (1) in the BTL model only depends on the difference of the reward functions, $r_\phi(y|\boldsymbol{x})$ admits an arbitrary constant shift, and thus we assume $r_\phi(y|\boldsymbol{x})$ to be normalized in a way such that

$$\mathbb{E}\left[p_{\text{base}}(y|\boldsymbol{x}) \exp\left(\frac{r_\phi(y|\boldsymbol{x})}{\beta}\right)\right] = 1,$$

which leads to the reparametrization $r_\theta(y|\boldsymbol{x}) = \beta \log \frac{p_\theta(y|\boldsymbol{x})}{p_{\text{base}}(y|\boldsymbol{x})}$, plugging which into the PPO loss (3) yields the DPO loss (4). □

*Proof of (10).* As in the derivation of the DPO loss (4) under the BTL model, we first consider the PPO algorithm for the PL model:

*Step 1.* Find the optimal score function $s_\phi(\boldsymbol{y}|\boldsymbol{x})$ that minimizes the loss function:

$$-\mathcal{L}_{\text{ListNet}}(s_\theta; \mathcal{D}_{\text{LTR}}^j) = \mathbb{E}\left[\sum_{i=1}^{n} t(z_i^j) \log \left(\frac{\exp(s_\phi(\boldsymbol{y}_i|\boldsymbol{x}))}{\sum_{i'=1}^{n} \exp(s_\phi(\boldsymbol{y}_{i'}|\boldsymbol{x}))}\right)\right]; \qquad (16)$$

*Step 2.* Fine-tune the base model $s_{\text{base}}$ with the optimal score function $s_\phi$ by maximizing the expected score value while penalizing the KL divergence between the new model and the base model:

$$-\mathcal{L}(p_\theta; p_{\text{base}}, r_\phi, \beta) = \mathbb{E}\left[s_\phi(\boldsymbol{y}|\boldsymbol{x})\right] - \beta D_{\text{KL}}(p_\theta||p_{\text{base}}) = \mathbb{E}\left[s_\phi(\boldsymbol{y}|\boldsymbol{x}) - \beta \log \frac{p_\theta(\boldsymbol{y}|\boldsymbol{x})}{p_{\text{base}}(\boldsymbol{y}|\boldsymbol{x})}\right].$$

$$(17)$$

For the optimization problem in the second step (17), following the same procedure as in the proof of (4), we solve the optimal $p_\theta$ by letting the functional derivative of the loss w.r.t. the density function $p_\theta(y|\boldsymbol{x})$ vanish and obtain

$$p_\theta(\boldsymbol{y}|\boldsymbol{x}) \propto p_{\text{base}}(\boldsymbol{y}|\boldsymbol{x}) \exp\left( \frac{s_\phi(\boldsymbol{y}|\boldsymbol{x})}{\beta} \right). \tag{18}$$

By the assumption of the PL model and the ListNet loss, we have $p_\theta(\boldsymbol{y}|\boldsymbol{x})$ modeled as the top-1 probability of the PL model and thus related to the score function $s_\theta(\boldsymbol{y}|\boldsymbol{x})$ via

$$p_\theta(\boldsymbol{y}|\boldsymbol{x}) = \frac{\exp(s_\theta(\boldsymbol{y}|\boldsymbol{x}))}{\sum_{i'=1}^n \exp(s_\theta(\boldsymbol{y}_{i'}|\boldsymbol{x}))}.$$

Let $p_{\text{base}}(\boldsymbol{y}|\boldsymbol{x}) = \frac{\exp(s_{\text{base}}(\boldsymbol{y}|\boldsymbol{x}))}{\sum_{i'=1}^n \exp(s_{\text{base}}(\boldsymbol{y}_{i'}|\boldsymbol{x}))}$, (18) can be rewritten as

$$\exp(s_\theta(\boldsymbol{y}|\boldsymbol{x})) \propto \exp\left( s_{\text{base}}(\boldsymbol{y}|\boldsymbol{x}) + \beta s_\phi(\boldsymbol{y}|\boldsymbol{x}) \right),$$

*i.e.*

$$s_\theta(\boldsymbol{y}|\boldsymbol{x}) = s_{\text{base}}(\boldsymbol{y}|\boldsymbol{x}) + \beta s_\phi(\boldsymbol{y}|\boldsymbol{x}) + C,$$

where $C$ is a constant shift. By noticing that the softmax function in (16) is invariant to the constant shift of the score function $s_\phi(\boldsymbol{y}|\boldsymbol{x})$, we may choose certain normalization such that

$$s_\theta(\boldsymbol{y}|\boldsymbol{x}) = s_{\text{base}}(\boldsymbol{y}|\boldsymbol{x}) + \beta s_\phi(\boldsymbol{y}|\boldsymbol{x})$$

holds, plugging which into the loss (16) yields the reparametrized ListNet loss (10). $\qquad\square$

## A.2 PROOFS OF LINEAR TRANSFORMATION PROPERTY

*Proof of Proposition 3.1.* For clarity, we first remove the penalization, *i.e.* to consider the case where $\lambda = 0$.

Then the ICOS loss (14) is of the following form:

$$\mathcal{L}_{\text{ICOS}}(s_\theta(\cdot, \boldsymbol{w}|\boldsymbol{x}); s_{\text{base}}, \boldsymbol{\beta}, \mathcal{D}_{\text{MOFT}})$$
$$= \mathbb{E}_{\boldsymbol{w} \sim \text{Dir}(\boldsymbol{\alpha})} \left[ \mathcal{L}_{\text{ListNet}, \boldsymbol{w}}(s_\theta(\cdot, \boldsymbol{w}|\boldsymbol{x}); s_{\text{base}}, \boldsymbol{\beta}, \mathcal{D}_{\text{MOFT}}) \right]$$
$$= \mathbb{E}_{\boldsymbol{w} \sim \text{Dir}(\boldsymbol{\alpha})} \left[ \sum_{j=1}^m w_j \mathcal{L}_{\text{ListNet}}(s_\theta(\cdot, \boldsymbol{w}|\boldsymbol{x}); s_{\text{base}}, \boldsymbol{\beta}, \mathcal{D}_{\text{MOFT}}^j) \right]$$
$$= \mathbb{E}_{\boldsymbol{w} \sim \text{Dir}(\boldsymbol{\alpha})} \left[ \sum_{j=1}^m w_j \mathbb{E} \left[ \sum_{i=1}^n t(z_i^j) \log \left( \frac{\exp\left(\beta_j(s_\theta(\boldsymbol{y}_i, \boldsymbol{w}|\boldsymbol{x}) - s_{\text{base}}(\boldsymbol{y}_i, \boldsymbol{w}|\boldsymbol{x}))\right)}{\sum_{i'=1}^n \exp\left(\beta_j(s_\theta(\boldsymbol{y}_{i'}, \boldsymbol{w}|\boldsymbol{x}) - s_{\text{base}}(\boldsymbol{y}_{i'}, \boldsymbol{w}|\boldsymbol{x}))\right)} \right) \right] \right]$$
$$= \mathbb{E} \left[ \sum_{j=1}^m \sum_{i=1}^n w_j t(z_i^j) \log \left( \frac{\exp\left(\beta_j(s_\theta(\boldsymbol{y}_i, \boldsymbol{w}|\boldsymbol{x}) - s_{\text{base}}(\boldsymbol{y}_i, \boldsymbol{w}|\boldsymbol{x}))\right)}{\sum_{i'=1}^n \exp\left(\beta_j(s_\theta(\boldsymbol{y}_{i'}, \boldsymbol{w}|\boldsymbol{x}) - s_{\text{base}}(\boldsymbol{y}_{i'}, \boldsymbol{w}|\boldsymbol{x}))\right)} \right) \right],$$

where the expectation in the second to last equality is taken over the data distribution $\mathcal{D}_{\text{MOFT}}$, and the expectation in the last equality is taken over both the data distribution $\mathcal{D}_{\text{MOFT}}$ and the weight distribution $\text{Dir}(\boldsymbol{\alpha})$ as a shorthand notation.

By the definition of the model $s_{\theta, \boldsymbol{\beta}}(\boldsymbol{y}, \boldsymbol{w}|\boldsymbol{x})$, we have that

$$s_{\theta, \boldsymbol{\beta}}(\boldsymbol{y}, \boldsymbol{w}|\boldsymbol{x}) = \arg\max_{s_\theta(\boldsymbol{y}, \boldsymbol{w}|\boldsymbol{x})} \mathbb{E} \left[ \sum_{j=1}^m \sum_{i=1}^n w_j t(z_i^j) \log \left( \frac{\exp\left(\beta_j(s_\theta(\boldsymbol{y}_i, \boldsymbol{w}|\boldsymbol{x}) - s_{\text{base}}(\boldsymbol{y}_i, \boldsymbol{w}|\boldsymbol{x}))\right)}{\sum_{i'=1}^n \exp\left(\beta_j(s_\theta(\boldsymbol{y}_{i'}, \boldsymbol{w}|\boldsymbol{x}) - s_{\text{base}}(\boldsymbol{y}_{i'}, \boldsymbol{w}|\boldsymbol{x}))\right)} \right) \right].$$

We now consider the following reparametrized optimization problem:

$$\max_{s_\theta'(\boldsymbol{y}, \boldsymbol{w}|\boldsymbol{x})} \mathbb{E} \Bigg[ \sum_{j=1}^m \sum_{i=1}^n w_j t(z_i^j)$$
$$\log \left( \frac{\exp\left(\beta_j(cs_\theta'(\boldsymbol{y}_i, \boldsymbol{w}|\boldsymbol{x}) + (1-c)s_{\text{base}}(\boldsymbol{y}_i, \boldsymbol{w}|\boldsymbol{x}) - s_{\text{base}}(\boldsymbol{y}_i, \boldsymbol{w}|\boldsymbol{x}))\right)}{\sum_{i'=1}^n \exp\left(\beta_j(cs_\theta'(\boldsymbol{y}, \boldsymbol{w}|\boldsymbol{x}) + (1-c)s_{\text{base}}(\boldsymbol{y}_{i'}, \boldsymbol{w}|\boldsymbol{x}) - s_{\text{base}}(\boldsymbol{y}_{i'}, \boldsymbol{w}|\boldsymbol{x}))\right)} \right) \Bigg]$$
$$= \mathbb{E} \left[ \sum_{j=1}^m \sum_{i=1}^n w_j t(z_i^j) \log \left( \frac{\exp\left(c\beta_j(s_\theta'(\boldsymbol{y}_i, \boldsymbol{w}|\boldsymbol{x}) - s_{\text{base}}(\boldsymbol{y}_i, \boldsymbol{w}|\boldsymbol{x}))\right)}{\sum_{i'=1}^n \exp\left(c\beta_j(s_\theta'(\boldsymbol{y}_{i'}, \boldsymbol{w}|\boldsymbol{x}) - s_{\text{base}}(\boldsymbol{y}_{i'}, \boldsymbol{w}|\boldsymbol{x}))\right)} \right) \right],$$

$$\tag{19}$$

obtained by reparametrizing $s_\theta(\boldsymbol{y}, \boldsymbol{w}|\boldsymbol{x})$ as

$$s_\theta(\boldsymbol{y}, \boldsymbol{w}|\boldsymbol{x}) = cs'_\theta(\boldsymbol{y}, \boldsymbol{w}|\boldsymbol{x}) + (1-c)s_{\text{base}}(\boldsymbol{y}, \boldsymbol{w}|\boldsymbol{x}), \tag{20}$$

and thus by solving

$$s_{\theta,\boldsymbol{\beta}}(\boldsymbol{y}, \boldsymbol{w}|\boldsymbol{x}) = cs'_\theta(\boldsymbol{y}, \boldsymbol{w}|\boldsymbol{x}) + (1-c)s_{\text{base}}(\boldsymbol{y}, \boldsymbol{w}|\boldsymbol{x}),$$

we have

$$s'_\theta(\boldsymbol{y}, \boldsymbol{w}|\boldsymbol{x}) = \frac{1}{c}s_{\theta,\boldsymbol{\beta}}(\boldsymbol{y}, \boldsymbol{w}|\boldsymbol{x}) - \frac{1-c}{c}s_{\text{base}}(\boldsymbol{y}, \boldsymbol{w}|\boldsymbol{x}) \tag{21}$$

is an optimal solution to the reparametrized optimization problem.

Notice that the function in the optimization problem (19) is exactly the ICOS loss (14) with the temperature $c\boldsymbol{\beta}$, we have that the $s_{\theta,c\boldsymbol{\beta}}(\boldsymbol{y}, \boldsymbol{w}|\boldsymbol{x})$ as defined in (15) coincides with the optimal solution (21). Thus we have proved the linear transformation property for the ICOS loss with $\lambda = 0$.

For the case with penalization, we assume the penalization term $\mathcal{G}_{\boldsymbol{w}}(s_\theta; s_{\text{base}}, \boldsymbol{\beta})$ is a function of the vector of ListNet losses $\boldsymbol{\mathcal{L}}_{\text{ListNet}}(s_\theta(\cdot, \boldsymbol{w}|\boldsymbol{x}); s_{\text{base}}, \boldsymbol{\beta}, \mathcal{D}_{\text{MOFT}})$, which is satisfied for the cosine similarity penalization loss (13) as proposed by Ruchte & Grabocka (2021). And in turn, the vector of ListNet losses $\boldsymbol{\mathcal{L}}_{\text{ListNet}}(s_\theta(\cdot, \boldsymbol{w}|\boldsymbol{x}); s_{\text{base}}, \boldsymbol{\beta}, \mathcal{D}_{\text{MOFT}})$ depends on $s_\theta(\cdot, \boldsymbol{w}|\boldsymbol{x})$ only in the form of $s_\theta(\cdot, \boldsymbol{w}|\boldsymbol{x}) - s_{\text{base}}(\cdot, \boldsymbol{w}|\boldsymbol{x})$, and therefore, we could write the ICOS loss in an abstract form as

$$s_{\theta,\boldsymbol{\beta}}(\boldsymbol{y}, \boldsymbol{w}|\boldsymbol{x}) = \underset{s_\theta(\boldsymbol{y}, \boldsymbol{w}|\boldsymbol{x})}{\arg\max} \, \mathbb{E}\left[\Phi\left(s_\theta(\cdot, \boldsymbol{w}|\boldsymbol{x}) - s_{\text{base}}(\cdot, \boldsymbol{w}|\boldsymbol{x})\right)\right],$$

*e.g.* for the case where $\lambda = 0$, $\Phi$ is of the following form:

$$\Phi(s_\theta(\cdot, \boldsymbol{w}|\boldsymbol{x}) - s_{\text{base}}(\cdot, \boldsymbol{w}|\boldsymbol{x}))$$
$$= \sum_{j=1}^{m}\sum_{i=1}^{n} w_j t(z_i^j) \log\left(\frac{\exp\left(\beta_j(s_\theta(\boldsymbol{y}_i, \boldsymbol{w}|\boldsymbol{x}) - s_{\text{base}}(\boldsymbol{y}_i, \boldsymbol{w}|\boldsymbol{x}))\right)}{\sum_{i'=1}^{n}\exp\left(\beta_j(s_\theta(\boldsymbol{y}_{i'}, \boldsymbol{w}|\boldsymbol{x}) - s_{\text{base}}(\boldsymbol{y}_{i'}, \boldsymbol{w}|\boldsymbol{x}))\right)}\right).$$

Apply the same reparametrization as in (20), we have that the reparametrized optimization problem is of the form

$$\max_{s'_\theta(\boldsymbol{y}, \boldsymbol{w}|\boldsymbol{x})} \mathbb{E}\left[\Phi\left(cs'_\theta(\cdot, \boldsymbol{w}|\boldsymbol{x}) + (1-c)s_{\text{base}}(\cdot, \boldsymbol{w}|\boldsymbol{x}) - s_{\text{base}}(\cdot, \boldsymbol{w}|\boldsymbol{x})\right)\right]$$
$$= \mathbb{E}\left[\Phi\left(cs'_\theta(\cdot, \boldsymbol{w}|\boldsymbol{x}) - cs_{\text{base}}(\cdot, \boldsymbol{w}|\boldsymbol{x})\right)\right],$$

with an optimal solution in the form of (21). Therefore, the linear transformation property also holds for the ICOS loss with the penalization term. $\qquad\square$

# B  ADDITIONAL EXPERIMENT DETAILS

In this section, we present additional details of the experiments conducted in the main text, including further descriptions of the baseline implementations, and the ablation studies of the HyperDPO framework.

## B.1  BASELINE IMPLEMENTATIONS

In the following, we will introduce and discuss the baseline methods used in the experiments in detail.

- *DPO Linear Scalarization (DPO-LS):* Given the base model $s_{\text{base}}$, for each weight vector $\boldsymbol{w} \in \mathbb{R}^m$, the DPO-LS method trains the new model $s_\theta$ with the loss function $\mathcal{L}_{\text{ListNet},\boldsymbol{w}}$ (12) and obtain $s_{\theta,\boldsymbol{w}}$ defined as

$$s_{\theta,\boldsymbol{w}} = \underset{s_\theta}{\arg\min} \, \mathcal{L}_{\text{ListNet},\boldsymbol{w}}(s_\theta; s_{\text{base}}, \boldsymbol{\beta}, \mathcal{D}_{\text{MOFT}})$$
$$= \underset{s_\theta}{\arg\min} \, \boldsymbol{w}^\top \boldsymbol{\mathcal{L}}_{\text{ListNet}}(s_\theta; s_{\text{base}}, \boldsymbol{\beta}, \mathcal{D}_{\text{MOFT}}).$$

This model is a naive generalization from the weighted sum method in the MOO literature to the MOFT problem, and the main drawback is that it needs as many training jobs and models as the number of sampled weight vectors, which is computationally expensive.

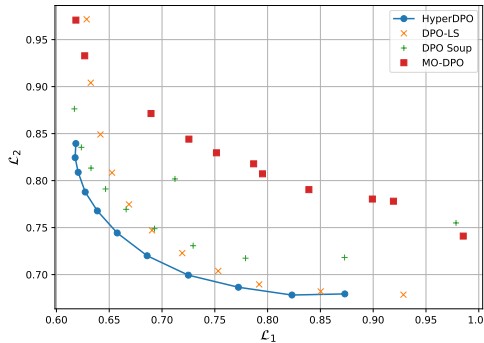

Figure 5: Comparison of Pareto fronts obtained by HyperDPO and the baselines on the PKU-SafeRLHF dataset with the GPT-2 model, including the MO-DPO method. The results for MO-DPO may not represent its best performance due to the possible conflict between the prompt tuning and the MO-DPO method.

- *DPO Soup (Rame et al., 2024):* The DPO Soup model first trains $m$ models $s_{\theta, \boldsymbol{e}_i}$ for each unit vector $\boldsymbol{e}_i$ in the $m$-dimensional space, *i.e.* $m$ DPO models w.r.t. the $m$ auxiliary objectives, respectively, and then linearly combines the $m$ models to obtain the final model with the weight vector $\boldsymbol{w}$ in the parameter space. The DPO Soup method offers a more efficient way to combine the models trained with different auxiliary objectives, but it still requires $m$ training jobs and models for each auxiliary objective, and the performance of this model is largely dependent on the landscape of the parameter space of the neural network architecture. As depicted in Figure 2, the Pareto front obtained by the DPO Soup method may present unexpected curves, and Figure 4 shows that the DPO Soup method may even exhibit mode collapse for certain combinations.

- *MO-DPO (Zhou et al., 2023):* The MO-DPO method also starts with the training of $m$ models $s_{\theta, \boldsymbol{e}_i}$ for each unit vector $\boldsymbol{e}_i$ in the $m$-dimensional space, and then instead of linearly combining the parameters, MO-DPO conducts a new training job for each weight vector $\boldsymbol{w} \in \mathbb{R}^m$ with the following loss function:

$$\mathcal{L}_{\text{MO-DPO}}(s_\theta; s_{\text{base}}, \boldsymbol{\beta}, \mathcal{D}_{\text{MOFT}}) = \mathbb{E}\left[\sum_{i=1}^{n} t(z_i^j) \log\left(\frac{\exp\left(\beta_j r_{\theta, \boldsymbol{w}}^{\text{MO-DPO}}\right)}{\sum_{i'=1}^{n} \exp\left(\beta_j r_{\theta, \boldsymbol{w}}^{\text{MO-DPO}}\right)}\right)\right],$$

where, for an arbitrary $i \in [m]$, $r_{\theta, \boldsymbol{w}}^{\text{MO-DPO}}$ is defined as

$$r_{\theta, \boldsymbol{w}}^{\text{MO-DPO}} := \frac{1}{w_i}\left(s_\theta(\boldsymbol{y}|\boldsymbol{x}) - s_{\text{base}}(\boldsymbol{y}|\boldsymbol{x}) - \sum_{i' \neq i} w_{i'}\left(s_{\theta, \boldsymbol{e}_i'}(\boldsymbol{y}|\boldsymbol{x}) - s_{\text{base}}(\boldsymbol{y}|\boldsymbol{x})\right)\right). \tag{22}$$

As MO-DPO requires $m$ training jobs and one addition training job for each weight vector, it may require more training time and computational resources compared to the DPO-LS and DPO Soup methods. For the LLM alignment task, we observe MO-DPO suffers from unstable training caused by the $1/w_i$ vector in the expression (22) especially when $w_i$ is close to zero, and exhibit less competitive performance. We suspect that the conflict between the prompt tuning and the MO-DPO method may lead to the suboptimal performance of MO-DPO in the LLM alignment task.

The HyperDPO framework is designed to address the limitations of the existing methods and provide a more efficient and effective way to profile the Pareto front of the MOFT problems.

### B.2 Ablation Studies

In this section, we provide the ablation studies of the HyperDPO framework, including the sensitivity of the concentration parameter $\boldsymbol{\alpha}$ in the Dirichlet distribution, the depth of the model, and the performance of two different NN parametrizations $s_{\theta, \boldsymbol{w}, \boldsymbol{\beta}}(\cdot, \cdot|\boldsymbol{x})$, namely (a) *Training from Scratch* and (b) *Augmentation Network*.



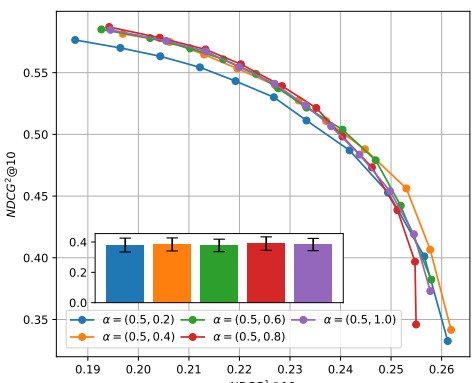

| $\alpha$ | Hypervolume |
|---|---|
| $(0.2, 0.2)$ | $1.446 \times 10^{-1}$ |
| $(0.4, 0.4)$ | $1.445 \times 10^{-1}$ |
| $(0.6, 0.6)$ | $1.471 \times 10^{-1}$ |
| $(0.8, 0.8)$ | $\mathbf{1.473 \times 10^{-1}}$ |
| $(1.0, 1.0)$ | $1.463 \times 10^{-1}$ |

(a) $\boldsymbol{\alpha} = (\alpha, \alpha)$ for $\alpha \in \{0.2, 0.4, 0.6, 0.8, 1.0\}$.

| $\alpha$ | Hypervolume |
|---|---|
| $(0.5, 0.2)$ | $1.451 \times 10^{-1}$ |
| $(0.5, 0.4)$ | $\mathbf{1.474 \times 10^{-1}}$ |
| $(0.5, 0.6)$ | $1.466 \times 10^{-1}$ |
| $(0.5, 0.8)$ | $1.458 \times 10^{-1}$ |
| $(0.5, 1.0)$ | $1.464 \times 10^{-1}$ |

(b) $\boldsymbol{\alpha} = (0.5, \alpha)$ for $\alpha \in \{0.2, 0.4, 0.6, 0.8, 1.0\}$.

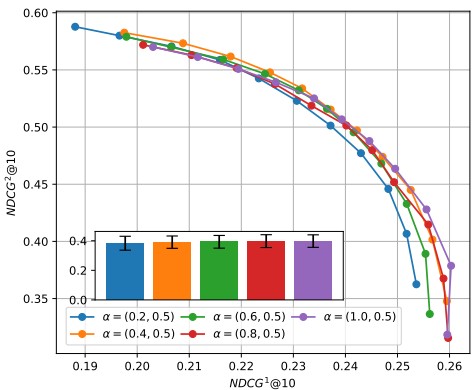

| $\alpha$ | Hypervolume |
|---|---|
| $(0.2, 0.5)$ | $1.447 \times 10^{-1}$ |
| $(0.4, 0.5)$ | $\mathbf{1.468 \times 10^{-1}}$ |
| $(0.6, 0.5)$ | $1.445 \times 10^{-1}$ |
| $(0.8, 0.5)$ | $1.444 \times 10^{-1}$ |
| $(1.0, 0.5)$ | $1.445 \times 10^{-1}$ |

(c) $\boldsymbol{\alpha} = (\alpha, 0.5)$ for $\alpha \in \{0.2, 0.4, 0.6, 0.8, 1.0\}$.

Figure 6: Ablation study on the impact of concentration parameter $\boldsymbol{\alpha}$ on the Pareto fronts obtained by the HyperDPO framework on the MSLR-WEB10K dataset (Objective I vs Objective II) with different settings of $\boldsymbol{\alpha}$. The hypervolume metric is shown in the table beside each figure.

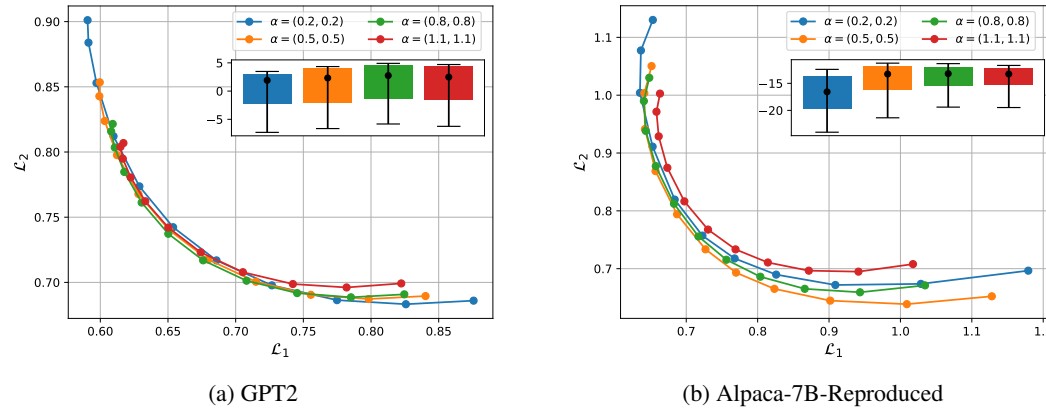

(a) GPT2                          (b) Alpaca-7B-Reproduced

Figure 7: Ablation study on the impact of the concentration parameter $\boldsymbol{\alpha}$ on the Pareto fronts obtained by the HyperDPO framework on the PKU-SafeRLHF dataset.

### B.2.1    CONCENTRATION PARAMETER $\boldsymbol{\alpha}$

The concentration parameter $\boldsymbol{\alpha}$ controls the span of the Dirichlet distribution from which the weight vector $\boldsymbol{w}$ is sampled and is the key parameter affecting the performance of the HyperDPO framework that should be carefully selected and validated. By the basic properties of the Dirichlet distribution, suppose $\boldsymbol{w} \sim \mathrm{Dir}(\boldsymbol{\alpha})$, then we have

$$\mathbb{E}[\boldsymbol{w}] = \frac{\boldsymbol{\alpha}}{\|\boldsymbol{\alpha}\|_1} := \overline{\boldsymbol{\alpha}}, \quad \mathrm{var}(\boldsymbol{w}) = \frac{\mathrm{diag}(\overline{\boldsymbol{\alpha}}) - \overline{\boldsymbol{\alpha}}\,\overline{\boldsymbol{\alpha}}^\top}{\|\boldsymbol{\alpha}\|_1 + 1}.$$

In general, the smaller the $\boldsymbol{\alpha}$, the more likely the weight vector $\boldsymbol{w}$ is close to the boundary of the simplex, and the larger the $\boldsymbol{\alpha}$, the more likely the weight vector $\boldsymbol{w}$ is concentrated around the expectation $\overline{\boldsymbol{\alpha}}$.

As the HyperDPO framework is generally robust to the choice of the concentration parameter $\boldsymbol{\alpha}$, we conduct ablation studies to investigate the impact of the concentration parameter $\boldsymbol{\alpha}$ on the performance of the HyperDPO framework in different settings. We first conduct experiments on the MSLR-WEB10K dataset with 2 auxiliary objectives (Query-URL Click Count vs URL Click Count) to investigate the impact of the concentration parameter $\boldsymbol{\alpha}$ on the performance of the HyperDPO framework. The results are shown in Figure 6. The experiment settings and plotting details are the same as in the main text.

As shown in Figure 6a, as the concentration parameter $\boldsymbol{\alpha}$ decreases, HyperDPO obtains a visually more comprehensive Pareto front thanks to more samples close to the boundary of the simplex. However, it is at the cost of a slightly undertrained model across the simplex, indicated by a lower hypervolume metric. It turns out that the choice of $\boldsymbol{\alpha}$ faces a trade-off between the diversity of the samples and the overall quality of the fine-tuning, given a fixed training budget. Similar trade-offs are observed in Figure 6b and 6c when only one dimension of the concentration parameter $\boldsymbol{\alpha}$ is varied.

We also conducted experiments on the PKU-SafeRLHF dataset to investigate the impact of the concentration parameter $\boldsymbol{\alpha}$ on the performance of the HyperDPO framework on the LLM alignment task. The results are shown in Figure 7. A similar pattern is observed in this large-scale task, where a smaller choice of the concentration parameter $\boldsymbol{\alpha}$ leads to a more comprehensive Pareto front. However, it does not necessarily lead to a worse hypervolume metric, suggesting that the performance of HyperDPO here is less hindered by the expressive power of the model, which has already been abundant in the LLM, but rather by the diversity of the samples.

### B.2.2    MODEL DEPTH

The depth of the neural network architecture is also crucial for the performance of the HyperDPO framework, as it determines the complexity and the expressiveness of the model. We also use the MSLR-WEB10K dataset with 2 auxiliary objectives (Query-URL Click Count vs URL Click Count)

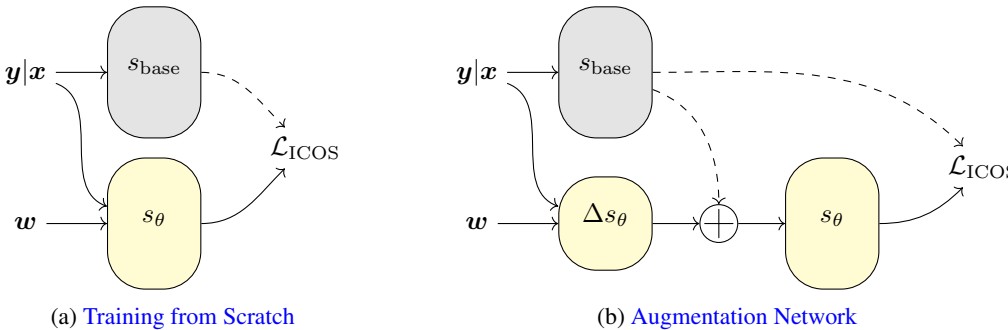

(a) Training from Scratch  (b) Augmentation Network

Figure 8: Illustration of two different parametrizations of the model $s_\theta(\cdot, \boldsymbol{w}|\boldsymbol{x})$ in the HyperDPO framework. Dashed lines denote that backpropagation is not applied.

to investigate the impact of *the model depth* on the performance of the HyperDPO framework. The results are shown in Figure 9a, where the depth, referring to the number of transformer layers in the model, is varied from 1 to 5. As shown in the figure, the performance of the HyperDPO framework is first significantly improved and gradually saturated with the increase of the depth. Besides, while the hypervolume metric improves, the coverage of the Pareto front does not change significantly with the increase in the depth. This suggests that the concentration parameter $\boldsymbol{\alpha}$ may have a more significant impact on the diversity of the samples than the model depth.

### B.2.3 MODEL PARAMETRIZATION

In general, one could adopt one of the two different parametrizations of $s_\theta(\cdot, \boldsymbol{w}|\boldsymbol{x})$ in the HyperDPO framework.

- *Training from Scratch*: The model $s_\theta(\cdot, \boldsymbol{w}|\boldsymbol{x})$ is a completely separate neural network from the base model $s_{\text{base}}(\boldsymbol{y}|\boldsymbol{x})$. Depending on the specific design of the for additional inputs $\boldsymbol{w}$, the new model may or may not share the same architecture as the base model. The main advantage of this design is that it requires less memory and computation resources (Rafailov et al., 2024b), and thus is more suitable for large-scale applications, *e.g.* LLMs.

- *Augmentation Network*: As several works (Chen et al., 2024; Xu et al., 2024) argue that DPO is prone to overfitting, one may curb the complexity of the model for the score function $s_\theta(\cdot, \boldsymbol{w}|\boldsymbol{x})$ by only adding a first-order correction term to the base model $s_{\text{base}}(\boldsymbol{y}|\boldsymbol{x})$ as:

$$s_\theta(\boldsymbol{y}, \boldsymbol{w}|\boldsymbol{x}) = s_{\text{base}}(\boldsymbol{y}|\boldsymbol{x}) + \Delta s_\theta(\boldsymbol{y}, \boldsymbol{w}|\boldsymbol{x}),$$

where the parameters in the base model are fixed, and the importance-conditioned design is only applied to the correction term $\Delta s_\theta(\cdot, \boldsymbol{w}|\boldsymbol{x})$. This design allows limited modification and reversibility to the base model and is thus suitable for applications where the fine-tuning is limited in budget, frequent, or expected to be minor.

The two parametrizations are illustrated in Figure 8a and 8b, respectively.

Both parametrizations can be seamlessly applied to the HyperDPO framework and easily switch between each other. In all the experiments presented in the main text, we have adopted the training from scratch design for the HyperDPO framework. Figure 9b shows the results of the HyperDPO framework with the augmentation training design on the same task as the previous ablation studies. Compared with Figure 9a, the augmentation training achieves a roughly better performance than the training from scratch design with the same depth, coinciding with the intuition that the augmentation training benefited from the information provided by the base model and instead of learning the entire score function $s_\theta(\cdot, \boldsymbol{w}|\boldsymbol{x})$ from scratch, it only needs to learn the correction term $\Delta s_\theta(\cdot, \boldsymbol{w}|\boldsymbol{x})$. When the model depth is increased, the performance of the augmentation training is also improved, sharing the same trend as the training from scratch design.

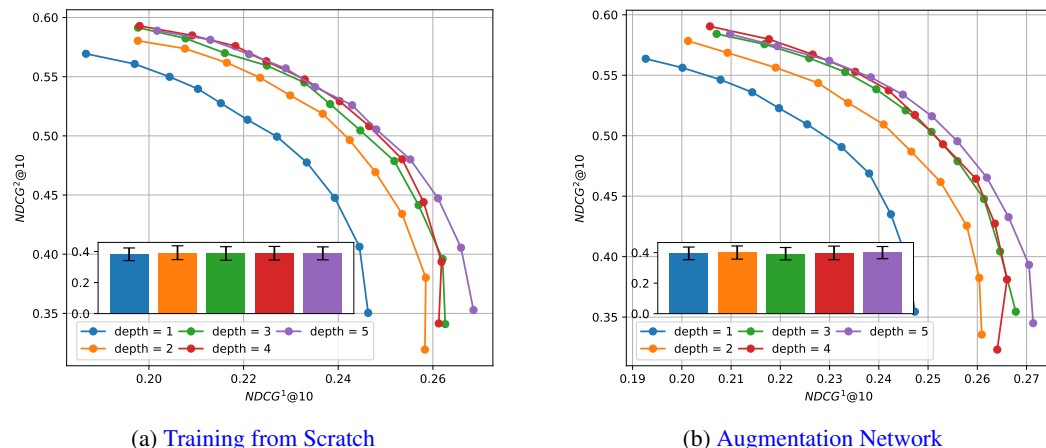

(a) Training from Scratch

(b) Augmentation Network

Figure 9: Ablation studies on the impact of the model depth and the model parametrizations on the Pareto fronts obtained by the HyperDPO framework on the MSLR-WEB10K dataset (Objective I vs Objective II).

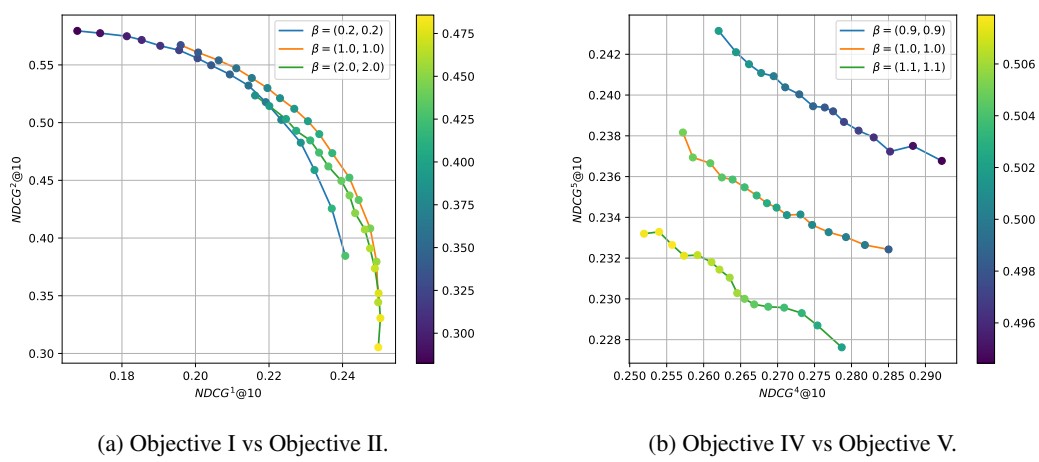

(a) Objective I vs Objective II.

(b) Objective IV vs Objective V.

Figure 10: Examples of post-training control over temperature $\beta$ on the MSLR-WEB10K dataset with 2 auxiliary objectives. Two axes denote the NDCG@10 of the two auxiliary objectives (the higher, the better). The colorbar denotes the NDCG@10 of the main objective.

## C  TOWARDS GENERALIZATION TO TEMPERATURE-CONDITIONED NETWORKS

In this section, we consider further generalization of the conditioned one-shot training technique to the temperature parameter $\beta$. Generally speaking, the model should exhibit different Pareto fronts for different temperature parameters $\beta \in \mathbb{R}_+^m$. By considering further conditioning on the temperature parameter $\beta$, we aim to output one score for each document $y$, denoted by $s_\theta(y, w, \beta | x)$, which reflects not only our importance weight $w$ between different auxiliary objectives but also the trade-off between the main objective and the auxiliary objectives controlled by the vector $\beta$.

### C.1  CURRENT POST-TRAINING CONTROL OVER TEMPERATURE $\beta$

Before we proceed to the temperature-conditioned networks, we would first present the current available post-training control over the temperature $\beta$ in the HyperDPO framework. As discussed in Section 3.4 after Proposition 3.1, the linear transformation property implies that the model can be scaled proportionally by a constant factor $c$ by a simple linear transformation of the output scores.

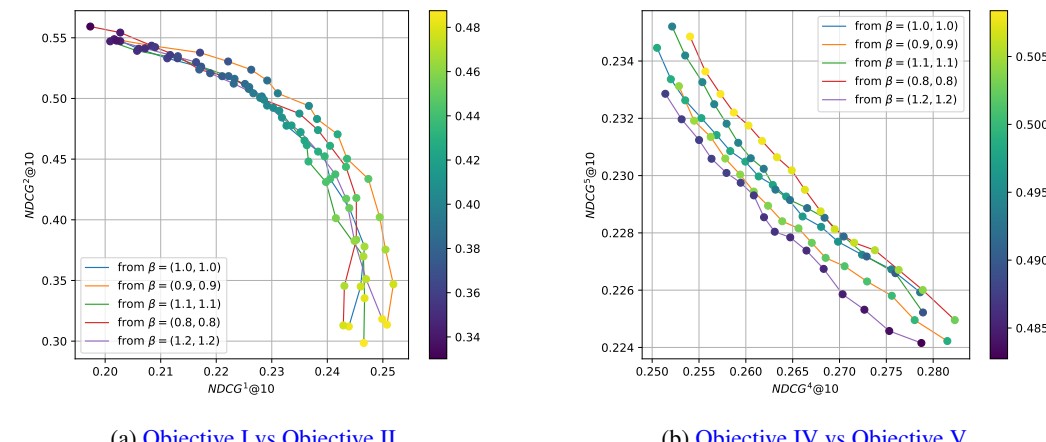

(a) Objective I vs Objective II.  (b) Objective IV vs Objective V.

Figure 11: Empirical validation of the linear transformation property on the MSLR-WEB10K dataset with 2 auxiliary objectives. The Pareto fronts in the figures are obtained by first training a model with the temperature $\boldsymbol{\beta}$ in the legend and then transform to the same temperature $\boldsymbol{\beta} = (1, 1)$ using post-training controls. Two axes denote the NDCG@10 of the two auxiliary objectives (the higher, the better). The colorbar denotes the NDCG@10 of the main objective.

Figure 10 gives examples of the post-training control over the temperature $\boldsymbol{\beta}$ on the MSLR-WEB10K dataset with 2 auxiliary objectives. As the temperature $\boldsymbol{\beta}$ increases, the Pareto front shifts towards the direction where the main objective is more emphasized, which is consistent with our expectations. In Figure 10b, the two auxiliary objectives are in balance, and thus, the shifts of the Pareto fronts resemble that depicted in Figure 1. However, in Figure 10a, the unexpected shifting pattern is observed, which may reflect the complex interactions between the main and auxiliary objectives.

Figure 11 provides empirical validation of the linear transformation property on the MSLR-WEB10K dataset with 2 auxiliary objectives. The methodology is that we first train an importance-conditioned network with the different temperatures $\boldsymbol{\beta}$ ranging from $(0.8, 0.8)$ to $(1.2, 1.2)$, and then transform the Pareto fronts obtained by the trained models to the same temperature $\boldsymbol{\beta} = (1, 1)$ using the post-training control as indicatd in (15). The penalization coefficient $\lambda$ is set to $0.05$ in the training. The results show that the transformed Pareto fronts are roughly aligned with each other, which validates the linear transformation property of the model. The slight deviation may be caused by the noises in the training process and the non-uniqueness of the optimal solutions of the ICOS loss.

Motivated by the observation of complicated trade-offs between the main and auxiliary objectives, one may consider using different temperature $\beta$ for different objectives and also a disproportionate post-training scaling of the temperature parameter $\boldsymbol{\beta}$ to achieve more flexible control over the Pareto front. To this end, we propose to design the *temperature-conditioned networks* to achieve this goal by incorporating the temperature parameter $\boldsymbol{\beta}$ into the input in a similar manner as the weight vector $\boldsymbol{w}$.

## C.2 MODEL PARAMETRIZATION

Proposition 3.1 implies that the temperature $\boldsymbol{\beta} \in \mathbb{R}_+^m$ actually has $m - 1$ degrees of freedom, and thus we propose to use the following reparametrization by projecting $\boldsymbol{\beta}$ to its $L^1$-normalization $\overline{\boldsymbol{\beta}} := \boldsymbol{\beta}/\|\boldsymbol{\beta}\|_1 \in \Delta^m$, *i.e.*

$$s_\theta(\boldsymbol{y}, \boldsymbol{w}, \boldsymbol{\beta}|\boldsymbol{x}) = \left(1 - \frac{1}{\|\boldsymbol{\beta}\|_1}\right) s_{\text{base}}(\boldsymbol{x}) + \frac{1}{\|\boldsymbol{\beta}\|_1} s_{\theta, \boldsymbol{w}, \overline{\boldsymbol{\beta}}}(\boldsymbol{x}). \tag{23}$$

Compared with importance-conditioned networks, the *temperature-conditioned networks* further take the temperature parameter $\boldsymbol{\beta}$ as input, and the model is trained to output the score for each

document $\boldsymbol{y}$ conditioned on both the importance weight vector $\boldsymbol{w}$ and the temperature parameter $\boldsymbol{\beta}$. The training is then conducted by randomly sampling $\boldsymbol{\beta} \in \mathbb{R}_+^m$ over a certain distribution $\mathcal{D}(\beta)$ valued in $\mathbb{R}_+^m$, and the loss of the *temperature-conditioned one-shot (TCOS) fine-tuning* can be written as

$$\mathcal{L}_{\text{TCOS}}(s_\theta; s_{\text{base}}, \mathcal{D}_{\text{MOFT}}, \boldsymbol{\alpha}, \lambda)$$

$$:= \mathbb{E}_{\boldsymbol{\beta} \sim \mathcal{D}(\boldsymbol{\beta})} \left[ \mathbb{E}_{\boldsymbol{w} \sim \text{Dir}(\boldsymbol{\alpha})} \left[ \mathcal{L}_{\text{ListNet}, \boldsymbol{w}}(s_\theta(\cdot, \boldsymbol{w}, \boldsymbol{\beta}|\boldsymbol{x}); s_{\text{base}}, \mathcal{D}_{\text{MOFT}}) + \lambda \mathcal{G}_{\boldsymbol{w}}(s_\theta(\cdot, \boldsymbol{w}, \boldsymbol{\beta}|\boldsymbol{x}); s_{\text{base}}) \right] \right].$$

The algorithm for the HyperDPO framework with TCOS fine-tuning is provided in Algorithm 2. The main difference between Algorithm 1 and Algorithm 2 is the additional sampling of the temperature parameter $\boldsymbol{\beta}$ and the reparametrization of the model $s_\theta(\cdot, \boldsymbol{w}, \boldsymbol{\beta}|\boldsymbol{x})$ as in Equation (23).

---

**Algorithm 2:** HyperDPO Framework with Temperature-Conditioned One-Shot Fine-Tuning

**Data:** Base model $s_{\text{base}}(\boldsymbol{y}|\boldsymbol{x})$, dataset $\mathcal{D}_{\text{MOFT}}$, concentration parameter $\boldsymbol{\alpha}$, penalization coefficient $\lambda$ (Training); temperature $\boldsymbol{\beta}$, weight vector $\boldsymbol{w}$ (Post-Training Control).

**Result:** Fine-Tuned Model $s_\theta(\cdot, \cdot, \cdot|\boldsymbol{x})$ (Training); $s_\theta(\boldsymbol{y}, \boldsymbol{w}, \boldsymbol{\beta}|\boldsymbol{x})$ (Post-Training Control).

    // Training
1 **for** $e = 1$ **to** $N_{\text{steps}}$ **do**
2      Sample $\boldsymbol{w}' \sim \text{Dir}(\boldsymbol{\alpha})$, $\boldsymbol{\beta}' \sim \mathcal{D}(\boldsymbol{\beta})$;
3      $\theta \leftarrow$
         $\theta - \eta \nabla_\theta \left[ \mathcal{L}_{\text{ListNet}, \boldsymbol{w}}(s_\theta(\cdot, \boldsymbol{w}', \boldsymbol{\beta}'|\boldsymbol{x}); s_{\text{base}}, \mathcal{D}_{\text{MOFT}}) + \lambda \mathcal{G}_{\boldsymbol{w}}(s_\theta(\cdot, \boldsymbol{w}', \boldsymbol{\beta}'|\boldsymbol{x}); s_{\text{base}}) \right]$;
4 **end**
    // Post-Training Control
5 $s_\theta(\boldsymbol{y}, \boldsymbol{w}, \boldsymbol{\beta}|\boldsymbol{x}) \leftarrow (1 - 1/c)\, s_{\text{base}}(\boldsymbol{y}|\boldsymbol{x}) + s_{\theta, \boldsymbol{\beta}}(\boldsymbol{y}, \boldsymbol{w}|\boldsymbol{x})/c$.

---

In general, the distribution $\mathcal{D}(\boldsymbol{\beta})$ should be chosen to cover a reasonable range of temperature parameters $\boldsymbol{\beta}$ to ensure the problem is tractable, as our experiments reveal that TCOS fine-tuning may require highly expressive neural networks to capture the complex trade-offs both between the main and auxiliary objectives and across the auxiliary objectives.

## C.3 PRELIMINARY RESULTS

All experiments presented in this section are conducted on the MSLR-WEB10K dataset with 2 auxiliary objectives (Quality Score vs Quality Score 2) to investigate the performance of the HyperDPO framework with TCOS fine-tuning, as it provides better visualization and comparisons of the Pareto fronts with different temperature parameters $\boldsymbol{\beta}$. In particular, we adopt the augmentation network design for temperature-conditioned networks for better expressive power and stability.

We provide the preliminary results of the HyperDPO framework with the TCOS fine-tuning on the LTR task in Figure 12. The model depth is chosen to be 5, and the distribution $\mathcal{D}(\boldsymbol{\beta})$ is set to be $\text{Unif}([0.67, 1.5]^2)$. The results demonstrate the TCOS fine-tuning is capable of capturing the trade-off between the main objective and the auxiliary objectives for all kinds of temperature configurations $\boldsymbol{\beta}$, and the Pareto fronts exhibit expected behaviors with different $\boldsymbol{\beta}$. These results suggest that temperature-conditioned one-shot fine-tuning is a promising direction for the HyperDPO framework to achieve more flexible control over the Pareto front.

Given the choices of the temperature parameters, the Pareto fronts in both Figure 10a and 10b should merge into one single point, which refers to the solution of the single-objective fine-tuning task with certain temperature parameter $\beta$. Although the results are roughly in accordance with the theoretical expectations, there are still small gaps that may be accounted for by the limit of the expressive power of the model and insufficient exploration over the weight vector $\boldsymbol{w}$.

To explain this, we present ablation studies to investigate the effect of the expressiveness of the model on the performance of the HyperDPO framework with TCOS fine-tuning. We applied models with 2 to 5 layers of transformer architecture, and the results show that the performance, indicated by the expected behaviors of the Pareto front, is drastically improved with the increase of the number of layers. While swallower models yield Pareto fronts with less expected behaviors and more noise, *e.g.* the concavity of the Pareto fronts in Figure 13b partially indicates the insufficiency of the training process, the model with 5 layers of transformer architecture in Figure 13d exhibits improved

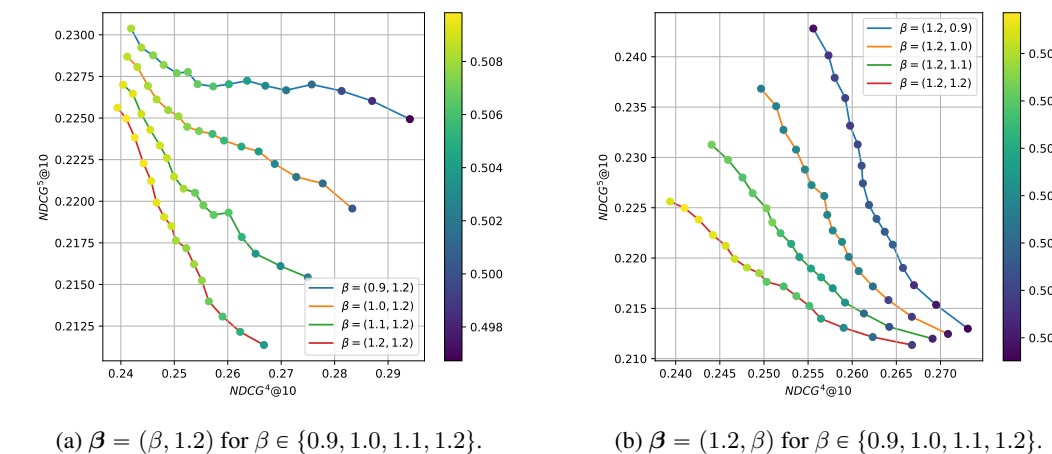

(a) $\boldsymbol{\beta} = (\beta, 1.2)$ for $\beta \in \{0.9, 1.0, 1.1, 1.2\}$.

(b) $\boldsymbol{\beta} = (1.2, \beta)$ for $\beta \in \{0.9, 1.0, 1.1, 1.2\}$.

Figure 12: Preliminary results of the HyperDPO framework with TCOS fine-tuning on the MSLR-WEB10K dataset (Objective IV vs Objective V). The colorbar denotes the NDCG@10 of the main objective.

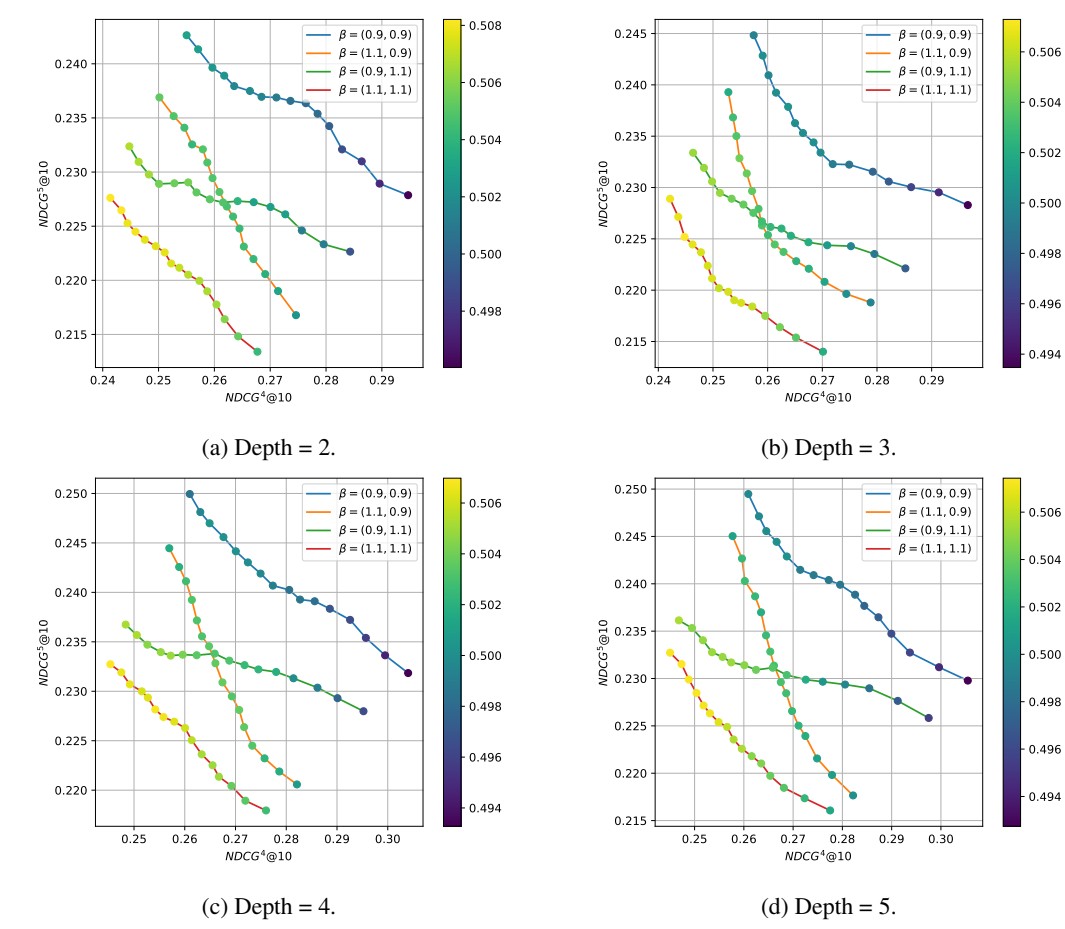

(a) Depth = 2.

(b) Depth = 3.

(c) Depth = 4.

(d) Depth = 5.

Figure 13: Ablation study of the impact of model depth on the Pareto fronts obtained by the Hyper-DPO framework with TCOS fine-tuning on the MSLR-WEB10K dataset (Objective IV vs Objective V). The colorbar denotes the NDCG@10 of the main objective.

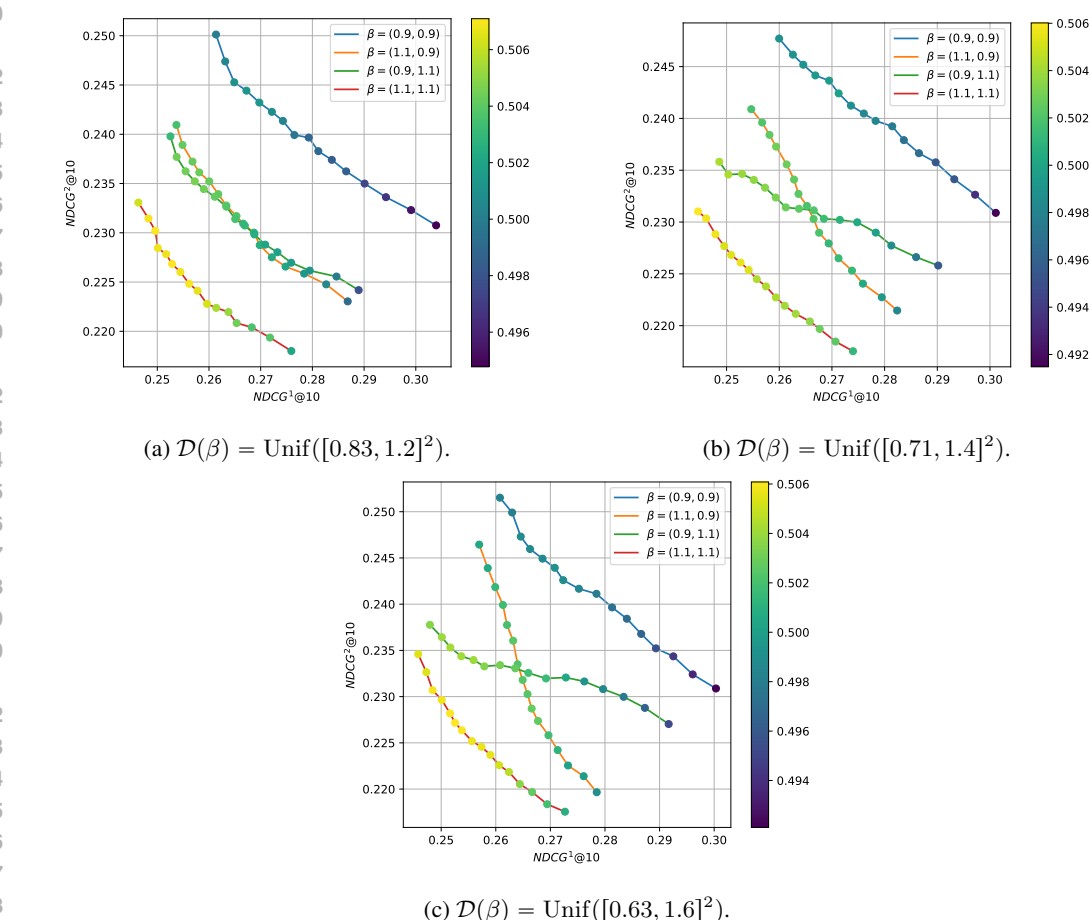

(a) $\mathcal{D}(\beta) = \mathrm{Unif}([0.83, 1.2]^2)$.

(b) $\mathcal{D}(\beta) = \mathrm{Unif}([0.71, 1.4]^2)$.

(c) $\mathcal{D}(\beta) = \mathrm{Unif}([0.63, 1.6]^2)$.

Figure 14: Ablation study of the impact of the distribution $\mathcal{D}(\beta)$ on the Pareto fronts obtained by the HyperDPO framework with TCOS fine-tuning on the MSLR-WEB10K dataset (Objective IV vs Objective V). The colorbar denotes the NDCG@10 of the main objective.

scores and more expected behaviors according to different temperature configurations. This suggests and confirms the intuition that TCOS fine-tuning require more expressive structures to capture the complex trade-offs between the main and auxiliary objectives.

The choice of the distribution $\mathcal{D}(\boldsymbol{\beta})$ also affects the performance of TCOS fine-tuning. Figure 14 shows the ablation study of the impact of the distribution $\mathcal{D}(\boldsymbol{\beta})$ on the Pareto fronts obtained by the HyperDPO framework with TCOS fine-tuning on the MSLR-WEB10K dataset. The results suggest that the distribution $\mathcal{D}(\boldsymbol{\beta})$ should cover a larger range than those interested to ensure sufficient training.

Given the preliminary results and ablation studies, we conclude that despite requiring more expressive structures and more training resources, temperature-conditioned one-shot fine-tuning is a feasible and promising direction for the HyperDPO framework to achieve more flexible control over the Pareto front and we expect to further investigate the validity of temperature-conditioned networks in future work.

