# OpenReview forum: "HyperDPO: Hypernetwork-based Multi-Objective Fine-Tuning Framework"
_ICLR.cc/2025/Conference — Submitted to ICLR 2025_

### Official Review · Reviewer_F9MP · 2024-10-30

**Soundness:** 2
**Presentation:** 2
**Contribution:** 2
**Rating:** 3
**Confidence:** 4

**Summary:**

The submission describes a system for multi-objective model finetuning, where a base model is trained on several new objectives from listwise ranking data. The main component is an input-conditioned network that takes additional inputs weights for the different objectives, and it trained with a random mixture over these. At prediction time, the weights can be set to achieve different trade-offs between objectives. Experiments for learning-to-rank and LLM-alignment show improved performance compared to baseline approaches. The supplemental material provides further (ablation) studies as well as an additional formulation in which not only the objective weights but also the temperature of their softmax can be adjusted at prediction time.

**Strengths:**

- timely and relevant topic

The submission addresses the largely unexplored problem of how LLMs can be fine-tuned in a multi-objective way, i.e. a network should be adapted with respect to not just one but multiple, potentially contradictory, targets and the trade-off between should be adjustable at prediction time. This matches well to real-world tasks, where LLMs should perform well in different metrics, e.g. utility, safety and fairness, and one cannot afford to train individual models for any desired trade-off.

- theoretical derivations as well as practical experiments

The submission presents mathematical derivations of their main components as well as experimental evaluations and ablation studies (but see below for shortcomings in presentation).

- experiments show improvement over baselines

The experimental evaluation shows that the proposed method tends to find Pareto-better solutions than baselines, and that it also takes less time for training (but again, see below).

**Weaknesses:**

While overall I believe the submission has merits, I found its presentation to leave several aspects unclear or even misleading.

1) the manuscript uses the term *hypernetwork* throughout the text, but the system it describes is not actually hypernetwork-based. A hypernetwork is a network that outputs the parameters of another network [Ha et al, "Hypernetworks", ICLR 2017], see also e.g. the recent survey [Chauhan et al, "A brief review of hypernetworks in deep learning", AI Review 2024]. In contrast, the system described in the submissions uses an input-conditioned network as in [Ruchte & Grabocka. "Scalable Pareto front approximation for deep multi-objective learning", ICDM 2021]. These two approaches are simply different, in their theoretical properties (see e.g. [Galanti & Wolf, "On the Modularity of Hypernetworks", NeurIPS 2020]) as well as in practice.

2) From the writing I was not able to extract the core technical contribution. Section 1.1 list the HyperDPO system as the main contribution, but that consists of several parts which existed (at least in similar form) previously. Does the main contribution lie in the differences? Or in the combination of techniques? Or in their application to the finetuning task?

To make my concern clearer: I believe for the reader it is most helpful if the "Preliminaries" section bundles prior work, and the "Methodology" section introduces the new scientific contribution on top of it. In the manuscript, the Methodology section indeed it introduces the multi-objective finetuning task, which I assume is a new formulation that can be seen as a generalization of standard LLM finetuning (thought as a special case of general multiobjective learing). But the Section discusses the classical Plackett-Luce model, and rederives the ListNet loss from [Cao et al, 2007]. Next, input-conditioned multi-objective learning [Ruchte & Grabocka, 2021] is introduced. Inserting the loss into the formulation and ultimately arrives at HyperDPO. It would be good to mark more clearly, which of the intermediate steps the authors consider scientific contributions and which are just provided for the convenience of the reader. The post-training control (via Proposition 3.1) appears new, but its relevance is not made clear here, and I have questions about the proof.

Other contribution mentioned in Section 1.1 are the Hyper Prompt Tuning, which however is first described in the experimental Section 4.2 at much lower level of detail than I would expect from a core contribution. The final contribution of a "temperature hypernetwork" is even only discussed in the supplemental material.

3) The experiments are only partially convincing.

The learning-to-rank experiments of Section 4.1 show promising results, but the settings is not the one used for motivating the method. The setup of the LLM alignment task is described rather briefly, e.g. the description of dataset, objectives and optimization process are too short to allow for reproducibility. The visualized Pareto curves and hypervolume values look promising, but absolute differences are small. In the LLM experiments, no error bars are provided, and generally statistical significance is not established or discussed. The training times would need more explanation or analysis as well. The times for the baselines as well as HyperDPO will depend on several factors, such as how many w-vector to sample and the total number of epochs. A justification is needed why the provided analysis is fair.

On a more fundamental level I am unsure what claim the experimental evaluation support (also related to 2). Section 4 compares the proposed input-conditioned network to simpler baselines, but [Ruchte & Grabocka, 2021] already did that (on different tasks, of course). Can the objective (13) be expected to be different from other MO tasks, so new evidence is required? Then experiments that highlight how (13) relates to alternative objectives would make the evidence stronger. If the multi-objective way of training is indeed the core contribution, I would have expected a comparison to other multi-objective methods from the literature, in particular actual multi-objective hypernetworks, such as [Navon et al, "Learning the Pareto Front with Hypernetworks", ICLR 2021] or [Hoang et al, "Improving Pareto Front Learning via Multi-Sample Hypernetworks", AAAI 2023].

(Please note that I do not ask for new experiments of this type to be reported in the author response)

Additional comments:
- the reference section has some issues: e.g. acronyms and conferences are often incorrectly capitalized, arXiv versions are cited where published version exist, some publications years seem incorrect (e.g. NeurIPS 2024 proceedings don't exist yet).
- floating point notation, such as "1.473e-01", should be change to mathematical notation, "0.1473e"

**Questions:**

The concerns I list in "weaknesses" make it hard to hard to judge the overall contribution and soundness of the work. Consequently, I gave low scores for these and the overall rating so far. I plan to revisit the scores after the author response.

* "hypernetwork"

I find *hypernetwork* not the correct term for the method you describe, given its use in the literature so far (see above). If you agree, please edit the manuscript accordingly. Otherwise, in case you have a different definition of hypernetwork in mind, please provide a reference for this.

* Contribution

Could you please clarify which aspects/parts of HyperDPO exactly you consider new scientific contributions? Or is it more of a systems-type contribution, where the novelty lies in the combination of existing components?

* Experiments

Related to "Contribution": what is the exact scientific claim for which the experiments provide evidence?

* Proposition 3.1

I have problems understanding the proof of Proposition 3.1 (page 16). Specifically, how does it follow from a comparison of $s_{\theta,c\beta}(y,w|x)$ with $s_{\theta,\beta}(y,w|x)$ that the numerators of both expressions have to be the same? Or do I misunderstand something else here?

---

> ### Author Response · Authors · 2024-11-19
> **Response to Reviewer F9MP (1/3)**
>
> We appreciate your comprehensive review and insightful comments. We are grateful for the opportunity to clarify the concerns raised regarding our manuscript.
>
> ---
>
> ## Regarding the Terminology "Hypernetwork"
>
> We acknowledge the confusion caused by our use of the term "hypernetwork" and appreciate the reviewer for pointing it out. In our manuscript, this term was intended to describe input-conditioned networks that directly output predictions, as utilized in the work by Ruchte & Grabocka [1], *i.e.* networks that are configured by hyperparameters. Our adaptation aimed to present these networks as an efficient alternative for the hypernetwork-based approaches for Pareto front profiling in multi-objective optimization contexts, as discussed in recent reviews [2].
>
> We seriously recognize the importance of aligning with established terminologies. To this end, we will revise the manuscript to replace "hypernetwork" with a more precise term, "importance/temperature-conditioned network," and ensure all references align with standard definitions. We would like to kindly refer the reviewer to our common response "Clarification on the Use of 'Hypernetwork'" as we explain this terminology misuse to all reviewers and provide our plan and commitment to revise the manuscript accordingly.
>
> ---
>
> ## Regarding the Core Technical Contribution
>
> We thank the reviewer for their comments on the core technical contributions of our work. We would kindly refer the reviewer to our response to the common concerns in this regard, where we have further elaborated on the motivation, formulation, and significance of our HyperDPO framework. We believe the core technical contributions of our work lie in not only the theoretical formulation of the multi-objective fine-tuning task and the HyperDPO methodology, but also the novel techniques, including the Hyper Prompt Tuning and the temperature-conditioned network, that are designed to address the challenges and demands of the real-world tasks. We have tried to prioritize the flow and the development of the MOFT task and the HyperDPO framework, and thus discuss the Hyper Prompt Tuning in the experiment section and the temperature-conditioned network in the supplemental material due to the space constraints.
>
> Following the reviewer's advice, we will revise the manuscript to better clarify the core technical contributions of our work, and mark those that are considered scientific contributions during the development of the HyperDPO framework clearly apart from those that are provided for the convenience of the reader. Most importantly, we will re-organize the manuscript to provide a more detailed discussion on our technical contributions, *i.e.* the Hyper Prompt Tuning and the temperature-conditioned network in the main text.

---

> ### Author Response · Authors · 2024-11-19
> **Response to Reviewer F9MP (2/3)**
>
> ## Regarding Experimental Evaluation
>
> We thank the reviewer for the comments on the experimental evaluation of our work, and we are glad to hear that the reviewer found the results "promising". We would like to further clarify the experimental evaluation and the scientific claims that the experiments provide evidence for:
>
> > The learning-to-rank experiments of Section 4.1 show promising results, but the settings is not the one used for motivating the method.
>
> We would like to point out that the LTR task does fit into the general multi-objective fine-tuning task, as the LTR task can be seen as a special case of the MOFT task, with an existing ranking model, and several additional datasets to fine-tune with. The LTR task is also more convenient to evaluate the performance of our HyperDPO framework, with straightforward evaluation of the performance w.r.t. the main objective. We believe our experimental results on the LTR task provide evidence for the effectiveness and efficiency of our HyperDPO framework in profiling the Pareto front and achieving a more comprehensive coverage of the Pareto front compared to the baselines.
>
> > The setup of the LLM alignment task is described rather briefly, e.g. the description of dataset, objectives and optimization process are too short to allow for reproducibility.
>
> We will expand the paragraph on the experimental setup in the LLM alignment task to provide more detailed descriptions of the dataset, objectives, and optimization process, to ensure the reproducibility of the results.
>
> > The visualized Pareto curves and hypervolume values look promising, but absolute differences are small.
>
> It is important to highlight that the performance of HyperDPO should be evaluated not only in terms of the visualized Pareto front curves and hypervolume metrics, but also training times, which are significantly reduced compared to the baselines. With a comparable and even better coverage, the training times of HyperDPO are more than **10x faster** than those of DPO-LS, which is a significant advantage for real-world applications.
>
> > In the LLM experiments, no error bars are provided, and generally statistical significance is not established or discussed.
>
> For the LLM alignment task, we have reported the deviation of the log-likelihood of the response from the reference model as a proxy of the deviation from the base model. The reported IQR of the log-likelihood deviation also indicates that the superior performance of HyperDPO is not due to a larger deviation from the base model. According to our current experiments, the results are consistent and robust across different random seeds. We will conduct more experiments with different random seeds to further validate the robustness of our results, as we strive to contribute reliable and reproducible results to the community and share our codebase and datasets for further validation.
>
> > The training times would need more explanation or analysis as well. The times for the baselines as well as HyperDPO will depend on several factors, such as how many w-vector to sample and the total number of epochs. A justification is needed why the provided analysis is fair.
>
> As for the ambiguity of the time complexity of the training of HyperDPO, we will provide better explanation of the training process in the main text to reduce confusion. In short, HyperDPO samples one preferenece weight vector $\boldsymbol w$ from the Dirichlet distribution before each epoch, evaluate the training loss using the current batch of data and the sampled $\boldsymbol w$, and conduct back-propagation. Consequently, the sampling process of $\boldsymbol w$ does not significantly increase the time complexity of the training process. The training times of HyperDPO are comparable to one training process w.r.t. one $\boldsymbol w$ value for the baselines, and this is also the reason why HyperDPO is approximately 10x faster than DPO-LS, when 11 $\boldsymbol w$ values are considered for the baselines.
>
> > Regarding comparisons with other baselines
>
> We acknowledge the reviewer's comments for the further comparison with other MOO methods from the literature. As we study the MOFT task, especially in the context of LLM alignment, we believe the comparison with the state-of-the-art baselines in the alignment literature is more relevant to the current development of the field. Also, as we have mentioned in the manuscript, additional technical and engineering considerations are needed to adapt the existing MOO methods to the MOFT task, which may of independent research significance and utility. While we believe the comparison with other MOO methods is a very important direction that definitely worths further exploration, it would be beyond the scope of the current work. We will further discuss future research directions and the potential extensions of our HyperDPO framework in the revised manuscript.

---

> ### Author Response · Authors · 2024-11-19
> **Response to Reviewer F9MP (3/3)**
>
> ## Regarding Other Comments
> - **Reference section**: We thank the reviewer for pointing out the issues. We will revise the reference section in the revised manuscript.
> - **Floating point notation**: We thank the reviewer for the suggestion. We will change the floating point notation to mathematical notation in the revised manuscript.
> - **Proposition 3.1**: We thank the reviwer for their insightful question. One should notice that $s_{\rm base}$ is given and fixed, and the function that is being optimized over is $s_{\theta}(\boldsymbol y_i, \boldsymbol w | \boldsymbol x)$.
>
> From the definition of $s_{\theta, \boldsymbol \beta}(\boldsymbol y, \boldsymbol w | \boldsymbol x)$, we have that $s_{\theta, \boldsymbol \beta}(\boldsymbol y, \boldsymbol w | \boldsymbol x) $ solves the following optimization problem:
> $$
> \max_{s_{\theta} (\boldsymbol y, \boldsymbol w | \boldsymbol x)} \mathbb{E}\left[ \sum_{i=1}^n t(z_i) \log \left(\dfrac{ \exp\big(\beta_j(s_{\theta}(\boldsymbol y_i, \boldsymbol w | \boldsymbol x) - s_{\rm base}(\boldsymbol y_i, \boldsymbol w | \boldsymbol x))\big)}{\sum_{i'=1}^n  \exp\big(\beta_j (s_{\theta}(\boldsymbol y_{i'}, \boldsymbol w | \boldsymbol x) - s_{\rm base}(\boldsymbol y_{i'}, \boldsymbol w | \boldsymbol x))\big)}\right) \right],
> $$
> and the argument in the original proof that $s_{\theta, c\boldsymbol \beta}(\boldsymbol y, \boldsymbol w | \boldsymbol x)$ solves
> $$
> \max_{s_{\theta}(\boldsymbol y, \boldsymbol w | \boldsymbol x)} \mathbb E\left[ \sum_{i=1}^n t(z_i) \log \left(\dfrac{ \exp\big(\beta_j (c s_{\theta}(\boldsymbol y_i, \boldsymbol w | \boldsymbol x) + (1 - c)s_{\rm base}(\boldsymbol y_i, \boldsymbol w | \boldsymbol x) - s_{\rm base}(\boldsymbol y_i, \boldsymbol w | \boldsymbol x))\big)}{\sum_{i'=1}^n \exp\big(\beta_j (c s_{\theta}(\boldsymbol y_{i'}, \boldsymbol w | \boldsymbol x) + (1 - c)s_{\rm base}(\boldsymbol y_{i'}, \boldsymbol w | \boldsymbol x) - s_{\rm base}(\boldsymbol y_{i'}, \boldsymbol w | \boldsymbol x))\big)}\right) \right].
> $$
>
> If we consider the optimization problem
> $$
> \max_s \mathbb E\left[ \sum_{i=1}^n t(z_i) \log \left(\dfrac{ \exp\big(\beta_j (s - s_{\rm base}(\boldsymbol y_i, \boldsymbol w | \boldsymbol x))\big)}{\sum_{i'=1}^n \exp\big(\beta_j (s - s_{\rm base}(\boldsymbol y_{i'}, \boldsymbol w | \boldsymbol x))\big)}\right) \right],
> $$
> then the two formulas above would yield two solutions $s_{\theta, \boldsymbol \beta}(\boldsymbol y, \boldsymbol w | \boldsymbol x)$ and $c s_{\theta, c\boldsymbol \beta}(\boldsymbol y, \boldsymbol w | \boldsymbol x) + (1 - c)s_{\rm base}(\boldsymbol y_i, \boldsymbol w | \boldsymbol x)$ that should actually match each other, by which we conclude that
> $$
> s_{\theta, c\boldsymbol \beta}(\boldsymbol y, \boldsymbol w | \boldsymbol x) = c s_{\theta, \boldsymbol \beta}(\boldsymbol y, \boldsymbol w | \boldsymbol x) + (1 - c)s_{\rm base}(\boldsymbol y_i, \boldsymbol w | \boldsymbol x).
> $$
>
>
> ---
>
> We would like to thank again the reviewer for their meticulous and constructive comments and suggestions. We hope the above clarifications and explanations address most of the concerns raised by the reviewer, and we have taken the feedback into account for the revision and betterment of our manuscript. We also hope the reviewer could reconsider the evaluation of our work based on our responses and revisions. Should the reviewer have any further questions or concerns, we are more than happy to engage in further discussions and integrate feedback into the revised manuscript.
>
>
> [1] Ruchte, Michael, and Josif Grabocka. "Scalable pareto front approximation for deep multi-objective learning." 2021 IEEE international conference on data mining (ICDM). IEEE, 2021.\
> [2] Hoang, Long P., et al. "Improving pareto front learning via multi-sample hypernetworks." Proceedings of the AAAI Conference on Artificial Intelligence. Vol. 37. No. 7. 2023.

---

> > ### Comment · Reviewer_F9MP · 2024-11-21
> > **Follow-up to Proof of Prop 3.1**
> >
> > Thank you for the edits, but I am still not able to follow the proof in its current form. While I do see the form of the two optimization problem, of course, the reasoning "compared with the definition" (in the PDF) or "should actually match each other" (above) is not clear to me. Also, because the argmax is not unique (at least up to constant shifts), one should to be careful to infer statements of "the" solution from the objective.
> >
> > Personally, I would suggest to at least invert the argument: after stating the transformed optimization problem, put forward $s \equiv \frac{1}{c}s_{\theta, \boldsymbol \beta}(\boldsymbol y, \boldsymbol w | \boldsymbol x) - \frac{1 - c}{c}s_{\rm base}(\boldsymbol y_i, \boldsymbol w | \boldsymbol x)$ as *a* candidate for an optimizer. When inserting this in the expression and simplifying, things end up in the same shape as before, and the optimality of the candidate follows from the fact that the resulting objective is minimal by definition of $s_{\theta, \boldsymbol \beta}$.

---

> ### Author Response · Authors · 2024-11-22
> **Reply to Comment on Proposition 3.1**
>
> Thank you for your thoughtful feedback and for highlighting the unclear aspects in our proof. We appreciate your suggestion regarding the restructuring of the argument. We have revised the statement of the linear transformation property in Proposition 3.1 in the main text as well as its proof accordingly, incorporating your approach to clearly delineate the steps and ensure the logical flow. This revised structure helps in demonstrating the optimality of the candidate more transparently and addresses the concerns regarding the uniqueness of the argmax. We hope that this modification improves the clarity and comprehensibility of the proof.

---

> > ### Author Response · Authors · 2024-11-23
> >
> > Thank you for your insightful feedback which has been instrumental in refining our manuscript. Following your suggestions, we have restructured the proof of Proposition 3.1 to more transparently demonstrate the optimality of the proposed solution, carefully addressing the uniqueness concerns related to the argmax. Additionally, we conducted and have now included new experiments in Figure 11 specifically designed to validate Proposition 3.1, which corroborate the theoretical adjustments made.
> >
> > Moreover, we have corrected the terminology "hypernetwork" to align with standard definitions, clarified our technical contributions, and enhanced our experimental evaluations to more robustly demonstrate the efficacy of our framework. These comprehensive revisions have been made with the aim to address the concerns raised during the review process.
> >
> > We believe these substantial updates significantly strengthen our manuscript and hope they will prompt a reconsideration of your evaluation. We are grateful for your guidance and remain open to further enhancing our work based on your expert feedback.

---

> > > ### Comment · Reviewer_F9MP · 2024-11-24
> > >
> > > Thank you for the edits and uploading the revised manuscript. I appreciate that you made substantial changes, including acknowledging that the method is not actually a hypernetwork, and your response did clarify some of my questions.
> > > However, after reading the revision and the other reviews I believe that the work would need a proper revision in order to qualify for a top publication venue. As I wrote in my original review, I do consider the formulation of the MOFT a conceptual contribution, though not a major one, as it mainly consists of existing components. As technical contribution, that I also had asked about, I do see that getting the HyperDPO (better called CondDPO?) to work is not a trivial feat. In term of method, the Hyper Prompt Tuning and the temperature-conditioned network are noteworthy. They might be sufficient to justify publication at a top conference, but that would require a description and experimental evaluation that is adjusted to them. Currently, the hyper prompt tuning is mentioned briefly in the experimental section, and the temperature network only in the appendix. The experiments are not set up to support them. Currently, they demonstrate that the conditioning-based multi-objective learning works better or faster than baseline methods, in the context of fine-tuning. This is useful, but prior work showed this in other tasks, so the new insight is limited. Instead, the evaluation should demonstrate that the actually newly proposed components are better than alternatives. In particular, if Hyper Prompt Tuning is a novel and relevant technique, I would like to see experiments comparing it to prior work or alternatives. Results about the temperature network are only in the appendix.
> > > Overall, I am not able to recommend acceptance for ICLR'24, but would encourage the authors to resubmit a revision to a future venue.

---

> ### Author Response · Authors · 2024-11-24
>
> Thank you for your detailed review and constructive suggestions that have significantly shaped the revisions of our manuscript. We are encouraged to hear that our formulation of the Multi-Objective Fine-Tuning (MOFT) problem and the HyperDPO (CondDPO) methodology are recognized for their relevance and non-trivial contributions. We also appreciate your acknowledgment of the novelty and potential impact of Hyper Prompt Tuning and the temperature-conditioned networks. The substantial revisions made, including terminology clarification, additional experiments, and reorganization of the text, have aimed to address your initial concerns comprehensively.
>
> ---
>
> > Currently, the hyper prompt tuning is mentioned briefly in the experimental section, [...]
>
> In response to your feedback, we have always attempted to strike a balance between comprehensiveness and readability, aiming to highlight both conceptual and methodological contributions without overshadowing the technical details. Regarding the Hyper Prompt Tuning, this technique represents a pioneering approach in the LLM alignment task, addressing unique challenges in the integration of continuous importance information into transformer-based models—marking a **first** in the literature. We have provided a detailed exposition of this technique in the experimental section to underscore its significance and utility, illustrated by Figure 3 and supported by the comparison with the traditional input augmentation design in the LTR task that highlight its necessity in the LLM alignment task.
>
> > [...] and the temperature network only in the appendix.
>
> For the temperature-conditioned networks, while we have expanded their discussion in the revised manuscript, we consciously chose to place detailed mathematical formulations and additional experimental results in the appendix to maintain the flow and accessibility of the main text. This decision was made considering the diverse expertise of our readership, ensuring that the paper remains focused and approachable while providing depth in supplementary sections for those interested in its potential generalization and more technical details.
>
> >  Instead, the evaluation should demonstrate that the actually newly proposed components are better than alternatives.
>
> We recognize the importance of further empirical comparisons and are committed to exploring these in future work. Currently, our focus has been to demonstrate the efficacy and innovation of our proposed methods within the established frameworks. Given the innovative nature of our approach and the extensive range of potential comparisons across the MOO and LTR literatures, conducting exhaustive empirical validation against every conceivable alternative would exceed the scope of a single seminal paper. Such extensive validation may constitute independent research interest and value, as it involves unique technical challenges that require additional advancements and research—similar to what we have achieved in our current work. For instance, we believe that further investigation into the mechanisms of Hyper Prompt Tuning and the development of potential alternatives present both relevant and significant research opportunities.
>
> ---
>
> In conclusion, we are confident that our manuscript significantly contributes to the field by presenting new theoretical insights, innovative methodologies, and robust experimental validations. Collectively, these elements enhance our understanding of MOFT problems, propose several promising future research directions, and affirm the suitability of our work for this prestigious venue. We hope that the revisions and further clarifications have effectively addressed your concerns and respectively request your kind reconsideration. Thank you once again for your time and invaluable feedback.

---

> > ### Author Response · Authors · 2024-11-26
> >
> > Dear Reviewer F9MP,
> >
> > We noticed that your review assigned a score of 2 (Fair) for both Soundness and Presentation. Given the updates and improvements we have made to the manuscript, we would like to ask for your perspective on whether the work still lacks soundness and clarity.
> >
> > If you believe these issues persist, we kindly request additional feedback on specific areas where the manuscript can be improved further. Alternatively, we respectfully ask you to reconsider these scores in light of the revisions. If deemed reasonable, we also invite you to reflect the improvements in the manuscript by adjusting the overall score, if appropriate.
> >
> > Given the initial rating, we understand that the adjustment to the final score can be made without altering your overall stance on the work, and we deeply appreciate your time and effort in reviewing our paper.

---

> > > ### Comment · Reviewer_F9MP · 2024-11-26
> > >
> > > Dear authors,
> > >
> > > please respect my right to have a different opinion about your submission than you do. In my review and response I detailed what I believe is needed to improve the manuscript. In particular, that is a clear description of the actual scientific contributions and experiments that support the main claims. These are not edits that could be made ad-hoc in a rebuttal, and once done they would have to be properly reviewed. As such, I keep my scores at below acceptance level.
> > >
> > > -- Reviewer F9MP

---

### Official Review · Reviewer_NRFT · 2024-11-03

**Soundness:** 4
**Presentation:** 4
**Contribution:** 3
**Rating:** 8
**Confidence:** 3

**Summary:**

This paper aims to tackle the multi-objective fine-tuning (MOFT) problem, where LLMs are trained towards satisfying multiple objectives simultaneously. As a main contribution, a method named HyperDPO is proposed. HyperDPO extends the Direct Preference Optimization (DPO) method with a hypernetwork that take weights of objectives and the input data. In this work, authors assumes the total utility (under MOO setting) is calculated via linear scalarization.  To demonstrate the effectiveness of HyperDPO, authors select three baseline methods, DPO-LS, DPO Soup, and MO-DPO. The authors select the hypervolume (HV) as the evaluation metric to compare the Pareto front. Experiments are conducted on two tasks, namely learning to rank and LLM alignment. Authors reported the results of fine-tuning GPT-2 and Alpaca-7B. The results demonstrate that HyperDPO could obtain higher HV with less training time, compared with baseline methods.

**Strengths:**

This work propose the HyperDPO method to address the MOFT problem. The technical contribution of this work is to incorporate a hypernetwork (through Hyper Prompt Tuning (HPT)) to DPO, such that the weights of objectives can be considered when fine-tuning LLM. While both Hypernetwork and DPO have been recently studied, the integration of both methods, espcially, using weights as inputs to the hypernetwork, seems to be a noval integration. In terms of writing, this paper is clearly written. In addition, it illustrate the theoretical foundation with clear detailed mathematical proofs. As DPO is a efficient way to fine-tune LLM, while source code is not available at the moment, HyperDPO might be an efficient way to fine-tune LLM under multi-objective setting.

**Weaknesses:**

This paper is overall well-written, however, the results could be further enhanced to better support the claim.  The first issue is that the results only include the hypervolume metric as the evaluation of pareto fronts (PF). Other important metrics including sparsity is not reported. In authors' experiments, they fine-tune two models, GPT-2 and Alpaca-7B. While results demonstrate HyperDPO perform better for both experiments, results in Figure 4 show that the performance of evaluated methods seems to be related to the base model. For example, while the PF of DPO-LS on GPT-2 is dominated by the one of HyperDPO, DPO-LS on Alpaca-7B performs quite close with HyperDPO. Similar pattern also reflects on HV and IQR. Therefore, it is not certain whether hyperDPO is generally better than other methods on different LLMs. More comparisons and analysis on when HyperDPO is a better method are expected.

In terms of citations, the inline citations could be improved. Generally, "(author, year)" in a sentence could not be used as a word.

**Questions:**

Here are my questions that requires further clarification from authors.
1. Why the results of MO-DPO are not reported in the Table 2?
2. How many times/random seeds that the experiments are conducted? In the results, HV and training time are not reported with STD, while the average main score is reported with STD, why here is a difference?
3. Can HyperDPO be extended to non-linear utility functions in the context of MOO?

---

> ### Author Response · Authors · 2024-11-19
> **Response to Reviewer NRFT (1/2)**
>
> We thank the reviewer for their positive comments and constructive feedback. We are glad to hear that the reviewer found our paper "*well-written*", theoretical foundation "*clearly detailed with mathematical proofs,*" and our proposed framework HyperDPO "*an efficient way to fine-tune LLM under multi-objective setting.*" In the following, we would like to address the concerns raised by the reviewer in a point-by-point manner.
>
> ---
>
> ## Regarding the Evaluation Metrics
>
> We thank the reviewer for the suggestion on including more evaluation metrics to better support the claim. As we are studying the fine-tuning task in a multi-objective setting, while we are aware of the importance of different evaluation metrics, our adoption of the hypervolume metric is motivated by the need to evaluate the effectiveness of our HyperDPO framework in profiling the Pareto front, as the hypervolume metric not only reflects the comprehensiveness of the coverage of the Pareto front, but also the performance of the model along each directions of the importance weights across auxiliary objectives, indicated by lower losses or higher scores. The hypervolume metric is also widely used in the MOO literature [1, 2]. We are interested in exploring the performance of our HyperDPO framework in terms of other evaluation metrics, such as sparsity, and their implications on the algorithm design and the real-world applications in the future work.
>
> ---
>
> ## Regarding the Performance of HyperDPO
>
> We understand the reviewer's concerns regarding how our experimental results support the claim that HyperDPO is generally better than other multi-objective fine-tuning methods. We would like to further clarify the interpretation of our experimental results and their implication on the performanc e of HyperDPO:
> - We are aware of that the performance of fine-tuning may be closely related to the extent that the new model is deviated from the base model. In general, one may trade off the performance on the main objective for extra performance gains on the auxiliary objectives. Although it is challenging to control this trade-off in an exact manner, we have used the same temperature $\beta$ across different multi-objective fine-tuning methods in our experiments, to ensure a fair comparison. As detailed in Appendix B.1, a common choice of the temperature $\beta$ generally leads to the same trade-off between the main and auxiliary objectives, and thus the performance of the methods should be comparable.
> - We have conducted experiments to validate the claim above on the LTR task, where the performance of the fine-tuned model w.r.t the main objective can be evaluated easily. As shown in Figure 2, we observe that the performance of the fine-tuned model on the main objective is generally consistent across different methods, while the performance on the auxiliary objectives varies. Furthermore, HyperDPO does achieve a more comprehensive and better coverage of the Pareto front compared to the baselines, as indicated by the hypervolume metric in Table 1.
> - For the LLM alignment task, since it is more challenging to evaluate the performance of the fine-tuned model on the main objective (which is generally a vague concept during pre-training), we propose to report the deviation of the log-likelihood of the response from the reference model as a proxy of the deviation from the base model. The reported IQR of the log-likelihood deviation also indicates that the superior performance of HyperDPO is not due to a larger deviation from the base model.
> - We would also like to point out that the performance of HyperDPO should also be taken account in terms of the training times, which are significantly reduced compared to the baselines.  As raised by the reviewer, the performance of HyperDPO shows slightly better than DPO-LS in terms of the hypervolume metric. Since the hypernetwork loss is designed to be a ''meta-learning'' problem that can recover the DPO-LS loss for specific $\boldsymbol w$ values, it is no wonder that the performance of HyperDPO is comparable to that of DPO-LS, and even better due to better generalization. However, the training times of HyperDPO are more than **10x faster** than those of DPO-LS, which is a significant advantage for real-world applications.
>
> ---
>
> ## Regarding the Citation Formats
>
> We thank the reviewer for pointing out the issue. We will revise the citation formats in the revised manuscript.

---

> > ### Comment · Reviewer_NRFT · 2024-12-02
> >
> > I would like to thank the authors for their responses to my questions.
> >
> > - While I agree with the authors that hypervolume is an important metric for evaluating the coverage of the Pareto fronts, there are additional important metrics too. For example, the sparsity measures the distributions of the non-dominated solutions and is widely considered in the domain of MOO too. As this work studies a problem of MOO, it is still suggested to include more results in evaluating the Pareto fronts. These results will be the baseline that will be compared with new methods.
> >
> > - For your response to the performance of HyperDPO, please refer to my comments on your response (2/2).

---

> ### Author Response · Authors · 2024-11-19
> **Response to Reviewer NRFT (2/2)**
>
> ## Regarding the Questions
> > Why the results of MO-DPO are not reported in the Table 2?
>
> We thank the reviewer for pointing out this issue. We kindly refer the reviewer to Figure 5, where the results of MO-DPO on the LLM alignment task with the GPT-2 model are reported and compared with the other methods. We conclude that the seemingly less favorable performance of MO-DPO therein may be possibly due to the possible conflict between the prompt tuning technique which Hyper Prompt Tuning is based on and the MO-DPO method, and our implementation of MO-DPO may represent its best performance. Given possible time-consuming process of the adaption of MO-DPO into our framework, considering the comparisons with MO-DPO in the LTR task in Figure 2 and Table 1, we choose to focus on the comparison with the other methods in the LLM alignment task.
> We will revise the manuscript to better clarify the results of MO-DPO and its comparison with other methods.
>
>
> > How many times/random seeds that the experiments are conducted? In the results, HV and training time are not reported with STD, while the average main score is reported with STD, why here is a difference?
>
> We acknowledge the reviewer's concerns regarding the number of times/random seeds that the experiments are conducted and the reporting of the standard deviations. For Figure 2 and Figure 3, all the experiments are conducted with the same random seed and the datasets are shuffled beforehand also with the same random seed to ensure the reproducibility of the results. Also, as far as we observed in our current scope of experiments, the results are consistent and robust across different random seeds. We will conduct more experiments with different random seeds to further validate the robustness of our results, as we strive to contribute reliable and reproduceable results to the community before and share our codebase and datasets for further validation.
>
>
> > Can HyperDPO be extended to non-linear utility functions in the context of MOO?
>
> We thank the reviewer for their insightful question. We believe that our HyperDPO framework can be extended to non-linear utility functions in the context of MOO. The key idea behind the HyperDPO framework is to leverage the hypernetwork structure to profile the Pareto front, which is a general and flexible way to model the trade-offs between the main and auxiliary objectives. The hypernetwork structure can be easily adapted to model non-linear utility functions, by replacing the hypernetwork loss (*cf.* Equation 13) with a more complex and expressive loss function that captures the non-linear relationships between the main and auxiliary objectives.
>
> Also, we believe our attempts to introduce the temperature hypernetwork have partially respond to this possible extension, as the normalization (Equation 19) introduces a more complicated and non-linear transformation to the temperature vector $\boldsymbol \beta$. We observed in Appendix C.3, especially in Figure 12, that the capture of the non-linear relationships requires a more expressive and flexible hypernetwork structure. We will further discuss the potential extensions of our HyperDPO framework to non-linear utility functions in the revised manuscript.
>
> ---
>
> We thank the reviewer again for their insightful comments and look forward to enhancing our manuscript based on this valuable feedback.
>
> [1] Zitzler, Eckart, and Lothar Thiele. "Multiobjective evolutionary algorithms: a comparative case study and the strength Pareto approach." IEEE transactions on Evolutionary Computation 3.4 (1999): 257-271.\
> [2] Navon, Aviv, et al. "Learning the pareto front with hypernetworks." arXiv preprint arXiv:2010.04104 (2020).

---

> > ### Comment · Reviewer_NRFT · 2024-12-02
> >
> > My main concern regarding this work is whether the performance of HyperDPO is statistically significant. From the response of the authors, it seems that the experiments related to the Pareto front were only conducted once (please correct me if I am wrong). Considering the training time is not costly, these experiments with a number of random seeds (the more the better) should be conducted and reported. As replied by the authors, "Also, as far as we observed in our current scope of experiments, the results are consistent and robust across different random seeds." If so, these results should be reported.
> >
> > I am still curious why the avg. main score is reported with standard deviations, while the hypervolume and training time is not.

---

> > > ### Author Response · Authors · 2024-12-02
> > > **Reply to Further Comments**
> > >
> > > We sincerely thank you for your thoughtful feedback and suggestions for improving our work. We would like to address your concerns regarding the experimental evaluation and statistical significance of our results.
> > >
> > > **Regarding the Evaluation Metrics**: We appreciate your suggestion about incorporating additional metrics beyond hypervolume for evaluating Pareto front coverage. We agree that sparsity is indeed an important metric in multi-objective optimization (MOO), as it provides valuable insights into the distribution of non-dominated solutions. We will revise our discussion to better contextualize our results within the broader framework of MOO evaluation metrics, specifically following the sparsity evaluation methodology established in Xu et al. (ICML 2020) and Faris et al. (ALA Workshop 2024).
> > >
> > > **Regarding Experimental Setup and Statistical Significance**: We would like to clarify several points about our experimental methodology:
> > > - For the purpose of presentation in Table 1, we reported results from one experimental run where all methods used the same random seed. For each method, we used 11 randomly selected weight vectors ($\boldsymbol w$) to produce 11 points for hypervolume calculation. The average main score and standard deviation reported in the table reflect the statistics of these 11 points. We will make this experimental setup more explicit in the revised paper.
> > > - We acknowledge your valid concern about the need for multiple runs with different random seeds. While our preliminary observations suggested consistency across different random seeds, we agree that proper statistical validation is necessary. Given the extremely tight deadline of the rebuttal period, we are working intensively to conduct as many additional experimental runs with multiple random seeds as possible within this timeframe.
> > >
> > > We appreciate your attention to these important methodological details. We will be sure to incorporate comprehensive experimental results with multiple random seeds in the next revision of our paper.
> > >
> > > [1] Xu, Jie, et al. "Prediction-guided multi-objective reinforcement learning for continuous robot control." *International conference on machine learning*. PMLR, 2020.
> > > [2] Faris, Jonathan G., et al. "Pareto Front Training For Multi-Objective Symbolic Optimization." *The Sixteenth Workshop on Adaptive and Learning Agents*. 2024.

---

### Official Review · Reviewer_JnjV · 2024-11-03

**Soundness:** 3
**Presentation:** 3
**Contribution:** 3
**Rating:** 6
**Confidence:** 3

**Summary:**

This paper focuses on Direct Preference Optimization (DPO) with multiple objectives. On top of the Plackett-Luce (PL) model, it proposes a generalized DPO problem that considers the Pareto front of multiple objectives. Afterward, by applying the hyper-network techniques, the authors propose a corresponding optimization algorithm through one-shot training, and offer a post-training control over the temperature hypermeter.

**Strengths:**

1. This work extends the DPO problem into the multiple-objective setting, which is important for the alignment of LLMs.
2. The proposed method connects the learning-to-rank problem and the LLM alignment. This might be instructive for developing more effective alignment methods.

**Weaknesses:**

1. The main concern is that the contributions of this work might be overclaimed. After formulating the multi-objective DPO problem, this work uses Ruchte & Grabocka's algorithm to profile the Pareto front.
2. Another contribution of this work is Prop. 3.1, which shows a linear transformation property between the optimal solutions under different $\beta$. However, according to the proof, this property is based on $\mathcal{L}\_{ListNet}$ (Eq. (12)) instead of $\mathcal{L}\_{Hypernet}$
 (Eq. (13)). Therefore, such a property ignores the Pareto fronts.
3. As one of the main contributions, the linear transformation property should have been validated in the experiments. Besides, how to apply this property in real scenarios should be further discussed.

**Questions:**

Please refer to the weaknesses.

---

> ### Author Response · Authors · 2024-11-19
> **Response to Reviewer JnjV (1/2)**
>
> We sincerely appreciate your insightful comments and the opportunity to clarify aspects of our work. We are committed to addressing your concerns thoroughly.
>
> ---
>
> ## Regarding the Contributions of the Work
>
> We thank the reviewer for their comments on the contributions of our work. We would like to refer to our response to the common concerns raised by the reviewers regarding the core contributions of the work, with further elaboration on the contributions of our work. We believe our work, claimed as a general framework for multi-objective fine-tuning tasks, has actually provided a clear and solid foundation for future research in this area, introduced a scalable and efficient solution for profiling the Pareto front, and developed a set of novel techniques to address the challenges and demands of the real-world tasks.
>
> We also like to further clarify the contributions of our work in relation to the work of Ruchte & Grabocka [1]. As Ruchte & Grabocka [1] proposed Pareto front approximations for multi-task learning tasks, our work extends their framework to the multi-objective fine-tuning tasks, which are more challenging and demanding due to the nature of the tasks and the models involved. To address these challenges, our work has introduced several novel techniques that are of independent interest, utility, and significance to the field:
> - **Hyper Prompt Tuning**: We propose the Hyper Prompt Tuning technique that bridges the gap between the hypernetwork-based MOO and the LLM alignment tasks, by transforming the importance weight vector $\boldsymbol w$ into a prefix to the input sequence embedding, which allows for efficient and effective incorporation of the importance information into transformer models.
> - **Temperature Hypernetwork**: We introduce the concept of temperature hypernetworks that provide a more flexible and adaptive way to control the trade-offs between the main and auxiliary objectives, which is crucial for real-world scenarios where the preferences may vary and change over time.
>
> We will revise the manuscript to better clarify the contributions of our work, with a focus on the novel techniques and the unique experimental findings that demonstrate the effectiveness and efficiency of our HyperDPO framework, and thus set our work apart from the existing literature.
>
> ---
>
> ## Regarding the Linear Transformation Property
>
> We thank the reviewer for pointing out this issue in the proof of Proposition 3.1. For simplicity, let us assume $\lambda = 0$ and therefore drop the penalization term at the moment, and thus the hypernetwork loss $\mathcal L_{\rm Hypernet}$ is nothing but the expectation of the ListNet loss $\mathcal L_{\rm ListNet}$ over the importance weight vector $\boldsymbol w$. Therefore, we have
> $$
> \begin{aligned}
> \mathcal L_{\rm Hypernet} &= \mathbb E_{\boldsymbol w \sim \mathrm{Dir}(\boldsymbol \alpha)}[\mathcal L_{\rm ListNet, \boldsymbol w}]\\\\
> &= - \mathbb E_{\boldsymbol w \sim \mathrm{Dir}(\boldsymbol \alpha)}\left[  \sum_{j=1}^m  w_j  \mathbb E_{\boldsymbol x, \boldsymbol y_i, z_i\sim \mathcal D_{\rm MOFT}} \left[
>   \sum_{i=1}^n t(z_i^j) \log \left(\dfrac{ \exp\big(c\beta_j(s_{\theta}(\boldsymbol y_i, \boldsymbol w | \boldsymbol x) - s_{\rm base}(\boldsymbol y_i, \boldsymbol w | \boldsymbol x))\big)}{\sum_{i'=1}^n  \exp\big(c\beta_j (s_{\theta}(\boldsymbol y_{i'}, \boldsymbol w | \boldsymbol x) - s_{\rm base}(\boldsymbol y_{i'}, \boldsymbol w | \boldsymbol x))\big)}\right)
> \right]
> \right]\\\\
> &= - \mathbb E_{\boldsymbol x, \boldsymbol y_i, z_i, \boldsymbol w}\left[
>  \sum_{j=1}^m  \sum_{i=1}^n w_j   t(z_i^j) \log \left(\dfrac{ \exp\big(c\beta_j(s_{\theta}(\boldsymbol y_i, \boldsymbol w | \boldsymbol x) - s_{\rm base}(\boldsymbol y_i, \boldsymbol w | \boldsymbol x))\big)}{\sum_{i'=1}^n  \exp\big(c\beta_j (s_{\theta}(\boldsymbol y_{i'}, \boldsymbol w | \boldsymbol x) - s_{\rm base}(\boldsymbol y_{i'}, \boldsymbol w | \boldsymbol x))\big)}\right)
> \right],
> \end{aligned}
> $$
> in which the last expectation is taken over both the data distribution $\mathcal D_{\rm MOFT}$ and the importance weight vector $\boldsymbol w$ drawn from the Dirichlet distribution $\mathrm{Dir}(\boldsymbol \alpha)$. Following the arguments in the original proof, one could derive the linear transform property by noticing the following transformation:
> $$
> c(s_{\theta}(\boldsymbol y_i, \boldsymbol w | \boldsymbol x) - s_{\rm base}(\boldsymbol y_i, \boldsymbol w | \boldsymbol x)) = (c s_{\theta}(\boldsymbol y_i, \boldsymbol w | \boldsymbol x) + (1 - c)s_{\rm base}(\boldsymbol y_i, \boldsymbol w | \boldsymbol x)) - s_{\rm base}(\boldsymbol y_i, \boldsymbol w | \boldsymbol x)
> $$

---

> ### Author Response · Authors · 2024-11-19
> **Response to Reviewer JnjV (2/2)**
>
> Furthermore, when $\lambda > 0$, since the penality term $\mathcal G_{\boldsymbol w}$ is a function of $\mathcal L_{\rm ListNet}$, and thus also the appearance of $s_{\theta}(\boldsymbol y_i, \boldsymbol w | \boldsymbol x)$ is also in the form of $c(s_{\theta}(\boldsymbol y_i, \boldsymbol w | \boldsymbol x) - s_{\rm base}(\boldsymbol y_i, \boldsymbol w | \boldsymbol x))$, the linear transformation property still holds by a similar argument.
>
> Therefore, our arguments regarding this linear transformation property is actually derived using the hypernetwork loss $\mathcal L_{\rm Hypernet}$ rather than only the ListNet loss $\mathcal L_{\rm ListNet}$ and does consider the Pareto fronts. We will revise the proof of Proposition 3.1 in the revised manuscript to better clarify the derivation.
>
> ---
>
> ## Regarding the Application of the Linear Transformation Property
>
> We thank the reviewer for the suggestion on validating the linear transformation property in the experiments and discussing its application in real scenarios. We would like to point out that besides the formulation and mathematical proofs of this property, we have also tested by experiments and incorporated it into our algorithm designs. We would like to further elaborate as follows:
> - We have tested the linear transformation property for the LTR task on the MSLR-WEB10K dataset, as presented in Figure 10. For the scenario Objective IV vs Objective V (Figure 10(b)), we have observed clear linear transformations of the Pareto fronts under different $\beta$ values, which are consistent with the theoretical property and the conceptual illustration in Figure 1. However, for the scenario Objective I vs Objective II (Figure 10(a)), the linear transformation presents a more complex pattern, which we believe is due to the imbalance between the two auxiliary objectives, and thus the Pareto front moves towards the direction where the main objective is more emphasized. These findings generally validate the linear transformation property and also open up new research questions and directions for future work.
> - The temperature hypernetwork is largely motivated by and grounded in the linear transformation property. As we have discussed in Appendix C.2, we use the property to reduce the degree of freedom of the temperature hypernetwork by one (*cf.* Equation (19)). Intuitively, we parametrize the temperature hypernetwork by first normalizing the temperature vector $\boldsymbol \beta$ by linear transformation of the scale $\|\boldsymbol \beta\|_1$, evaluating the hypernetwork, and then transforming the output back to the original scale. According to our experiment experience, this design choice is crucial for the performance of the temperature hypernetwork, and several experimental results are presented in Figure 11, 12, and 13.
>
> We will further discuss the application of the linear transformation property in real scenarios and the implications of the temperature hypernetwork in the revised manuscript, to better clarify the significance and utility of this property and the techniques derived from it.
>
> ---
>
> We would like to thank again the reviewer for the comments and questions. We hope the above clarifications and explanations address the concerns raised by the reviewer, based on which the reviewer could possibly reconsider the evaluation of our work. Should the reviewer have any further questions or concerns, we would be more than happy to engage in further discussions and integrate feedback into the revised manuscript.
>
> [1] Ruchte, Michael, and Josif Grabocka. "Scalable pareto front approximation for deep multi-objective learning." 2021 IEEE international conference on data mining (ICDM). IEEE, 2021.

---

### Official Review · Reviewer_8Vxc · 2024-11-09

**Soundness:** 3
**Presentation:** 2
**Contribution:** 2
**Rating:** 5
**Confidence:** 3

**Summary:**

This work extends Direct Preference Optimization (DPO) to the multi-objective settings.
This is achieved by replacing the BTL model to PL model that can handle list-wise rankings.

The work also profiles the Pareto Front with hypernetworks. The Pareto Front is a set of solutions involving trade-offs such that no objective can be improved further without hurting other objectives.

**Strengths:**

Fine tuning with multiple objectives is an important problem with many real world applications such as alignment of LLMs with multiple goals in helpfulness, trust and safety etc.

**Weaknesses:**

The technical contribution of this work is quite incremental. It is a combination of known techniques.
- The generalization due to PL was originally proposed in Liu 2024.
- Using a hypernetwork to profile the Pareto Front was proposed in Navon 2020.

**Questions:**

In the experiment, how would you describe the significance of the improvements?

---

> ### Author Response · Authors · 2024-11-19
> **Response to Reviewer 8Vcx (1/2)**
>
> We thank the reviewer for their insightful comments and constructive feedback. We appreciate the opportunity to clarify and further detail the contributions and results of our work. We address each concern in a point-by-point manner.
>
> ---
>
> ## Regarding the Technical Contributions of the Work
>
> We acknowledge the reviewer's comments on the technical contributions of our work. For a comprehensive discussion on the unique and novel contributions, we refer to our detailed response in the common concerns section. We emphasize that our work innovatively formulates the multi-objective fine-tuning problem, introduces novel techniques to tackle its challenges, and achieves competitive results with our HyperDPO framework, underscoring its effectiveness and efficiency.
>
> Furthermore, we would like to remark on the reviewer's comments on the comparison of our work with Liu et al. [1] and Navon et al. [2]:
>
> - *Comparison with Liu et al. [1]*: We understand the reviewer's concerns regarding our modeling of ranking data using the Plackett-Luce model. We would like to clarify that our work do not aim to propose a new model for ranking data, but rather to leverage the Plackett-Luce model as a tool to model the preference data in the context of multi-objective fine-tuning. The Plackett-Luce model is a well-established model and its application to LTR originates from Cao et al. [3]. To the best of our knowledge, Liu et al. [1] aims to make use of ranking datasets in the context of DPO, while our work focuses on the generalization of the DPO methodology to multi-objective settings. Furthermore, we are particularly careful about the implications of the Plackett-Luce model to the profiling of the Pareto fronts, as studied in Proposition 3.1 and the experiments, which we believe are novel and significant contributions of our work.
> - *Comparison with Navon et al. [2]*: We acknowledge the reviewer's comments on the use of hypernetworks for profiling the Pareto front, which may seem akin to that proposed in Navon et al. [2]. We would like to clarify that our work is distinct from Navon et al. [2] in several aspects.
>     - Firstly, Navon et al. designs a separate neural network that takes in the importance weight vector $\boldsymbol w$ and outputs the *parameters* of the neural network that takes in the input data and outputs its prediction. In contrast, our hypernetwork is designed to take in both the input data and the importance weight vector $\boldsymbol w$ and outputs the *predictions* directly. This design choice is especially crucial for the LLM Alignment task, as it allows us to efficiently incorporate the importance information into the model without altering the model architecture.
>     - Secondly, Navon et al. [2] focuses on the learning of the Pareto front using hypernetworks, while our work is centered around the fine-tuning in a multi-objective setting, which poses distinct challenges and demands. Our work not only leverages the hypernetwork structure for profiling the Pareto front, but also introduces several novel techniques, such as the Hyper Prompt Tuning and the Temperature Hypernetwork, that further enhance the applicability and effectiveness of hypernetworks.
>
> In conclusion, we believe our HyperDPO is rather a novel and significant contribution to the field of multi-objective fine-tuning, with a set of unique and innovative techniques and promising experimental results that render our work distinct from the existing literature [1, 2] and of general interest and utility to the community. We will revise the manuscript to better clarify the contributions of our work and the distinctions from the existing literature.

---

> ### Author Response · Authors · 2024-11-19
> **Response to Reviewer 8Vcx (2/2)**
>
> ## Regarding the Significance of the Improvements in the Experiments
>
> Thank you for highlighting the need for clarity on the experimental improvements. Our HyperDPO framework significantly enhances Pareto front coverage, reduces training times, and offers greater hyperparameter flexibility:
>
> - **Coverage and Smoothness**: As visualized by Figure 2 and 4, and measured with the hypervolume metric in Table 1 and 2, our HyperDPO framework is able to achieve a more comprehensive coverage of the Pareto front compared to the baselines in both the LTR and LLM alignment tasks. We also observe that the improvements are consistent across tasks with Pareto fronts of different geometry (*i.e.* convex (Figure 1(a)) vs non-convex (Figre 1(b))), and tasks with middle-sized (GPT-2, Figure 3(a)) vs large-sized (Alpaca-7B, Figure 3(b)) models. Furthermore, the Pareto fronts obtained by our HyperDPO framework are smoother and less prone to overfitting.
> - **Efficient in Training**: As shown in Table 1 and 2, our HyperDPO framework is able to achieve competitive performance with significantly reduced training times compared to the baselines. We would like to point out that for all previous methods, one need at least one extra model and training process for each $\boldsymbol w$ value, while our HyperDPO framework is able to achieve the same performance with a **single model and one-shot training process**. This is especially crucial for real-world applications where training times are a critical factor.
> - **Adaptability**: We have also conducted abundant experiments to demonstrate the flexibility and adaptability of our HyperDPO framework to different hyperparameters.
>   - HyperDPO is shown to be robust to the choice of the concentration parameter $\boldsymbol \alpha$ in the Dirichlet distribution, while as the model depth increases, the performance of HyperDPO improves consistently.
>   - We have also proposed and tested two different parametrizations, *i.e.* training-from-sketch and augmentation training, confirming the validity and discussing the suitability of each parametrization.
>   - As a first in the literature, we introduce the concept of temperature hypernetwork, which allows for a more flexible and adaptive control over the trade-offs between the main and auxiliary objectives. As shown in the experimental results in Appendix C.3, temperature hypernetworks are shown to achieve superior flexibility and performance compared to the ordinary weight hypernetworks.
>
> These improvements are substantiated by rigorous testing and are pivotal for real-world applications, where efficiency and adaptability are crucial.
>
>
> ---
>
> We hope our responses adequately address the reviewer’s concerns and demonstrate the depth and impact of our contributions. In light of the clarifications and additional details provided, we kindly ask the reviewers to reconsider the evaluation of our manuscript. We remain open to further discussion and are committed to integrating additional feedback to enhance our work. Thank you once again for your valuable insights and for considering our request.
>
> [1] Liu, Tianqi, et al. "Lipo: Listwise preference optimization through learning-to-rank." arXiv preprint arXiv:2402.01878 (2024).\
> [2] Navon, Aviv, et al. "Learning the pareto front with hypernetworks." arXiv preprint arXiv:2010.04104 (2020).\
> [3] Cao, Zhe, et al. "Learning to rank: from pairwise approach to listwise approach." Proceedings of the 24th international conference on Machine learning. 2007.

---

> > ### Author Response · Authors · 2024-11-23
> >
> > Thank you again for your constructive feedback. We hope our responses have adequately addressed your concerns. In response to your comments, we have significantly revised our theoretical framework, expanded our experimental validations, and corrected our use of terminology to align with standard definitions. Given these comprehensive changes, we kindly request that you reconsider your evaluation of our manuscript. We are committed to further refining our work and welcome any additional feedback you might have.

---

> ### Author Response · Authors · 2024-11-25
>
> As we approach the conclusion of this discussion phase, we kindly ask that you reevaluate our revised manuscript in light of the recent updates and enhancements we have made. We hope our efforts have addressed your concerns effectively. Thank you for your continued support and consideration.

---

### Author Response · Authors · 2024-11-19
**Common Responses (1/3)**

We sincerely thank all the reviewers for their constructive feedback and for recognizing the contributions of our work. We are heartened by the positive remarks highlighting our framework as "*timely and relevant,*" "*novel,*" "*clearly written,*" offering "*improved performance compared to baseline approaches,*" and that "*might be an efficient way to fine-tune LLM under multi-objective setting.*" Such endorsements affirm the value and potential impact of our research. Below, we address two common concerns raised by multiple reviewers, ensuring our commitment to clarity and precision in our scientific communication.

---

## Regarding the Core Contributions of the Work

We acknowledge the concerns raised by reviewers regarding the core contributions of the work. We would like to clarify the contributions of our work as follows:

### Formulation of the Multi-Objective Fine-Tuning (MOFT) Problem

Our newly-formulated MOFT problems are motivated by and designed to extract commonalities among several existing problems, such as Learning-to-Rank (LTR) [1], Multi-Objective Optimization (MOO) [2], and LLM alignment [3], especially Direct Preference Optimization (DPO) [4]. As scattering methods and techniques have been proposed for these problems, we point out that as long as the preference across different items according to one query are quantified in the form of logits, the problems and their fine-tuning can be unified under a single framework, with a general solution that can be applicable to a wide range of tasks and generalizable to multi-objective settings.

As commented by Reviewer F9MP, our work "*addresses the largely unexplored problem of how LLMs can be fine-tuned in a multi-objective way,*" and "*matches well to real-world tasks*". Given the pervasiveness of such problems where one would like to fine-tune an existing model that may be time-consuming to retrain with multiple objectives, we believe our formulation of the MOFT problem would serve as a novel, relevant, and solid foundation for future research in this area.

### Introduction of HyperDPO Framework

With the common characteristics and challenges among the problems clearly identified, we are subsequently able to adapt the state-of-the-art techniques from LTR, MOO, and LLM Alignment to the MOFT problem.

- The HyperDPO framework are grounded in the Plackett-Luce model for listwise ranking, powered by the hypernetwork structure for Pareto front profiling, and particularly designed for fine-tuning.
- As we mentioned in the manuscript, the HyperDPO framework can be seen as the generalization of LTR, MOO, and DPO problems, and there would be no wonder that our framework is capable of handling a wide range of tasks, as demonstrated by the superior performance on both LTR and LLM alignment in our experiments, which are of very distinct natures.
- We have also explored several theoretical properties of the HyperDPO framework in Proposition 3.1, so as to illustrate the rigorousness and rationale behind our framework.

As commented by Reviewer F9MP, the integration of hypernetwork and DPO "*seems to be a novel integration*" and by Reviewer JnjV, "*connects the learning-to-rank problem and the LLM alignment*", and "*might be instructive for developing more effective alignment methods*", we believe our hypernetwork-based method is a novel and useful way for MOFT problems, and we are excited to share our unique experimental findings with the community.

---

> ### Author Response · Authors · 2024-11-19
> **Common Responses (2/3)**
>
> ### Technical Contributions
>
> The engineering behind the HyperDPO framework are no easy feast. The HyperDPO framework is designed to be a general-purpose solution for MOFT problems, and it is not trivial to adapt the existing techniques to the MOFT problem without diminishing their generality and effectiveness. Notably, we introduced and developed the following techniques to address the challenges and demands of the MOFT problems:
>
> - *Hyper Prompt Tuning*: For example in the LLM alignment task, one generally may not be willing to alter the transformer architecture itself, but rather to convey the information of the importance weight $\boldsymbol w$ to the model in a more efficient and straightforward way. As a first among the literature, the Hyper Prompt Tuning technique is designed to achieve this goal, by transforming $\boldsymbol w$ into a prefix to the embedding of the input sequence, which by the properties of the transformer, aptly incorporates the importance information into the model, while minimizing the impact on the training process and the GPU-related issues.
> - *Temperature Hypernetwork*: We are also the first to propose the concept of temperature hypernetwork for MOFT problems, as we observe the need for a more flexible and adaptive way to control not only the trade-offs between the auxiliary objectives (achieved by the ordinary weight hypernetwork), but also those between the main and auxiliary objectives in real-world scenarios. The temperature hypernetwork, again, demands a new set of engineering techniques, which we have strived to develop and present, and our results are positive and promising.
>
>
> ### State-of-the-Art Experimental Results
>
> Based on the formulation of the MOFT problem and the HyperDPO framework and techniques, we have conducted a large set of experiments on two distinct tasks, namely the LTR and LLM alignment tasks. Our experimental results show that the HyperDPO framework has achieved the state-of-the-art performance in terms of efficient Pareto front profiling in the following aspects:
> - **Comprehensive Pareto Front Coverage**: Our HyperDPO framework is able to achieve a more comprehensive coverage of the Pareto front compared to the baselines in both the LTR and LLM alignment tasks, across tasks with Pareto fronts of different geometry, and tasks with middle-sized vs large-sized models.
> - **Efficient Training Times**: Featuring a simple one-shot training process, our HyperDPO framework is able to achieve competitive performance with significantly reduced training times compared to the baselines, which is especially crucial for real-world applications.
> - **Hyperparameter Robustness and Flexibility**: Ablation studies have proved the robustness of our HyperDPO framework to the choice of hyperparameters, observed the consistent performance improvement with the expressiveness of the hypernetwork, and demonstrated the superior flexibility and performance of the temperature hypernetworks compared to the ordinary weight hypernetworks.
>
> In conclusion, we believe our work is a novel, solid, and promising contribution to the field of multi-objective fine-tuning, with a set of unique and innovative techniques that are designed to address the challenges and demands of the real-world tasks. Our methods are corroborated by competitive experimental results showing state-of-the-art capabilities in Pareto front profiling with strikingly efficient training times on distinct tasks, which we believe would be of general interest and utility to the community.

---

> ### Author Response · Authors · 2024-11-19
> **Common Responses (3/3)**
>
> ## Clarification on the Use of "Hypernetwork"
>
> As raised by Reviewer F9MP, we acknowledge that our use of the term "hypernetwork" has led to confusion, as it deviates from the established definition predominantly recognized in the literature, such as defined in Ha et al. [5], where a hypernetwork specifically refers to a network that generates parameters for another network. In our manuscript, we referred to "hypernetworks" in the context of input-conditioned networks that directly output predictions, aligning more closely with the systems described by Ruchte & Grabocka [6]. We realize that this represents a departure from the conventional usage and could potentially mislead readers.
>
> In our responses to individual comments, we will continue using the original terminology for consistency and clarity in the context of our current discussion. However, we wish to assure you that we are committed to a thorough revision of our manuscript to rectify this terminology misuse. In the revised manuscript, we will:
>
> - Replace the term "hypernetwork" with "input-conditioned network" or other alternative appropriate term depending on the context that accurately reflects the functionality of our proposed system.
> - Ensure that all references and terminologies are aligned with current standard definitions as recognized within the field.
> - Provide clearer explanations of our methodologies and their alignment with established concepts.
>
> We believe that these revisions will not only correct the terminological errors but also enhance the manuscript's clarity and its alignment with the broader scientific discourse. We would want to emphasize that this change in terminology will not affect the effectiveness of our proposed framework. The conceptual and operational aspects of our framework remain robust and continue to deliver the demonstrated benefits in multi-objective fine-tuning.
>
> [1] Liu, Tie-Yan. "Learning to rank for information retrieval." Foundations and Trends® in Information Retrieval 3.3 (2009): 225-331.\
> [2] Sener, Ozan, and Vladlen Koltun. "Multi-task learning as multi-objective optimization." Advances in neural information processing systems 31 (2018).\
> [3] Christiano, Paul F., et al. "Deep reinforcement learning from human preferences." Advances in neural information processing systems 30 (2017).\
> [4] Rafailov, Rafael, et al. "Direct preference optimization: Your language model is secretly a reward model." Advances in Neural Information Processing Systems 36 (2024).\
> [5] Ha, David, Andrew Dai, and Quoc V. Le. "Hypernetworks." arXiv preprint arXiv:1609.09106 (2016).\
> [6] Ruchte, Michael, and Josif Grabocka. "Scalable pareto front approximation for deep multi-objective learning." 2021 IEEE international conference on data mining (ICDM). IEEE, 2021.

---

### Meta-Review · Area_Chair_r2Vv · 2024-12-18

**Metareview:**

This paper introduces a Multi-Objective Fine-Tuning framework named HyperDPO, which applies to various machine learning tasks, including Learning-to-Rank and aligning large language models (LLMs). Initially, the paper received mixed reviews. After the rebuttal phase, the final feedback remains inconsistent, with one positive (8), two borderline (6, 5), and one negative review (3). To make a final decision, I thoroughly read and discussed the comments with SAC, with special attention to the negative one. Overall, I agree with the three non-negative reviews. That is, it is a promising direction to integrate hypernetwork to DPO, and this paper is validated by detailed mathematical proofs. However, the shortcomings of this paper are also significant, and I decide to reject this paper due to the following reasons.

- As pointed out by Reviewer ``F9MP``, although the proposed method outperforms the baseline methods in the context of fine-tuning, prior work has shown similar performance gain in other tasks [Ruchte & Grabocka, 2021]. Hence,  more evidence is necessary to demonstrate that the gain comes from the newly proposed components. Otherwise, it is hard to evaluate the effectiveness of the proposed method.
- Some important concepts, such as hypernetwork, are confused. Specifically, hypernetwork has appeared in [Ha et al, ICLR 2017], which is inherently different from that proposed in this paper. It is recommended to replace the term with the other one, and the description of the proposed method should be modified accordingly. Hence, a significant revision is required.

To sum up, I recognize the value of HyperDPO (or CondDPO, according to the advice of Reviewer ``F9MP``). However, According to the careful and professional suggestions proposed by Reviewer ``F9MP``, I think a major revision is necessary to make this paper qualified for a top publication venue. I encourage the authors to revise these problems carefully and resubmit the paper to a future venue.

**Additional Comments On Reviewer Discussion:**

- Reviewer ``NRFT``: 8, very positive to the paper
- Reviewer ``JnjV``:  increased to 6, acknowledge the authors' reply
- Reviewer ``8Vxc``: 5,  mainly concerned about the contributions, did not engage in the rebuttal
- Reviewer ``F9MP``: 3, is still unconvinced of the effectiveness of the newly proposed components, and some of their contents require significant revision.

The reason for the decision is listed in the meta-review

---

### Decision · Program_Chairs · 2025-01-22

Reject